# Epidermal injury-induced derepression of key regulator *ATML1* in newly exposed cells elicits epidermis regeneration

Hiroyuki Iida [1,2], Ari Pekka Mähönen [2], Gerd Jürgens [3] & Shinobu Takada [1]✉

Plant cell fate determination depends on the relative positions of the cells in developing organisms. The shoot epidermis, the outermost cell layer of the above-ground organs in land plants, protects plants from environmental stresses. How the shoot epidermis is formed only from the outermost cells has remained unknown. Here we show that when inner leaf mesophyll cells are exposed to the surface, these cells show up-regulation of *ATML1*, a master regulator for epidermal cell identity in *Arabidopsis thaliana*. Epidermal cell types such as stomatal guard cells regenerate from young inner-lineage tissues that have a potential to accumulate ATML1 protein after epidermal injury. Surgical analyses indicate that application of pressure to the exposed site was sufficient to inhibit *ATML1* derepression in the outermost mesophyll cells, suggesting this process requires pressure release. Furthermore, pharmacological analyses suggest that *ATML1* derepression in the outermost mesophyll cells require cortical microtubule formation, MAPK signaling and proteasome activity. Our results suggest that surface-positional cues involving mechanical signaling are used to restrict *ATML1* activity to the outermost cells and facilitate epidermal differentiation.

Many plant cells maintain their ability to be reprogrammed and develop into different cell types. Specifically, they can acquire totipotency after exogenous phytohormone- or injury-induced dedifferentiation[1]. Cell ablation experiments in the meristem have shown that eliminated cell types are regenerated from invading neighboring cells of other identities, which suggests that plant cell fate can be respecified according to the position of the cell, regardless of its lineage, during regeneration[2–4]. The epidermis is one of the well-known cell types that are specified in a position-dependent manner. The shoot epidermis is formed from the outermost cell layer of land plants and is characterized by anticlinal cell division patterns, cuticle deposition and the presence of specialized cell types, such as stomata and trichomes[5,6]. More than 40 years ago, Stewart and Dermen showed that

when epidermal cells underwent unusual periclinal cell divisions, the outer daughter cells maintained epidermal cell identity and the inner daughter cells were differentiated into mesophyll cells[7]. This report suggests that the "outermost" cell position is essential for acquiring and maintaining epidermal cell identity and also implies that gene activities promoting epidermis or mesophyll fate might be changed in response to the displacement of the cells. However, the molecular mechanisms underlying the position-dependent cell fate change have remained unknown. In the last two decades, several regulators for epidermal cell identity have been identified in *Arabidopsis thaliana*. *ARABIDOPSIS THALIANA MERISTEM L1 LAYER* (*ATML1*), which encodes an HD-ZIP class IV transcription factor, is expressed preferentially in the outermost cells and promotes epidermal cell identity[8–11]

[1]Department of Biological Sciences, Graduate School of Science, Osaka University, 1-1 Machikaneyama, Toyonaka, Osaka 560-0043, Japan. [2]Organismal and Evolutionary Biology Research Programme, Faculty of Biological and Environmental Sciences, Viikki Plant Science Centre, University of Helsinki, Helsinki, Finland. [3]Center for Plant Molecular Biology (ZMBP), Developmental Genetics, University of Tübingen, Tübingen, Germany. ✉e-mail: shinobu_takada@bio.sci.osaka-u.ac.jp

(Supplementary Fig. 1a). Mutations in *ATML1* and its closest homologue, *PROTODERMAL FACTOR2* (*PDF2*), caused the formation of leaves lacking an epidermis[8]. In addition, constitutive expression of *ATML1* induced ectopic epidermis-related traits in the inner tissues of cotyledons and leaves[9,10]. These results suggest that ATML1 is a master transcription factor for epidermal cell identity. Therefore, it is expected that the outermost cell-specific activity of *ATML1* is responsible for the single epidermal layer formation in *Arabidopsis thaliana*. Indeed, we have reported that *ATML1* activity is restricted to the outermost cells by both transcriptional activation and post-transcriptional repression[12]. However, it is still unclear how transcription of *ATML1* in the outermost cells is established. Our previous report implied that cells should be located at the outermost positions to maintain *ATML1* activity because nuclear ATML1 accumulation was reduced in the inner daughter cells of the epidermis that underwent a rare periclinal division[12]. So far, however, it has not been experimentally addressed whether the outermost location of cells is sufficient to trigger *ATML1* expression or not. In this study, we surgically removed or injured the leaf epidermis and found that *ATML1* transcription was upregulated in the mesophyll cells exposed to the outermost position. We found that exogenous application of mechanical pressure represses *ATML1* upregulation in the mesophyll, suggesting that this process requires pressure release. In addition to the pressure release, cortical microtubule formation, MAPK cascade and proteasome activity were also necessary for the *ATML1* derepression in the mesophyll cells. Moreover, injury of outer tissues in young leaves induced epidermis regeneration from inner-lineage cells that can accumulate ATML1 protein.

We have provided molecular and mechanical bases to explain epidermal/mesophyll cell fate changes associated with changes in relative cell positions. Our results suggest that surface-positional cues involving mechanical signals facilitate transcriptional derepression of *ATML1* in the outermost cells to generate the epidermal barrier at the interface between plants and their environments to survive terrestrial conditions.

## Results

### *ATML1* expression was increased in mesophyll cells exposed to the surface position

To test the dependence of *ATML1* transcription on the cell position, we surgically removed the leaf epidermis and exposed subepidermal mesophyll cells to the outermost position. To observe *ATML1* transcriptional activity, we used *gATML1-nls-3xGFP* plants, in which triple GFP fusion protein with an SV40 nuclear localization signal (NLS) is produced under the control of the native *ATML1* regulatory sequence[12] (Supplementary Fig. 1b). Weak *gATML1-nls-3xGFP* signals were detected in mesophyll cells right after removal of the epidermis, which suggests that *ATML1* was weakly transcribed in mesophyll cells (Fig. 1a). One day after epidermis removal, GFP signals were increased in the mesophyll cells exposed to the surface position (Fig. 1b, m, n).

Next, we examined whether known regulatory sequences are sufficient for the *ATML1* upregulation in mesophyll cells. First, we utilized *proATML1-nls-3xGFP* plants, in which the *nls-3xGFP* reporter gene was expressed under the 3.4-kb *ATML1* promoter sequence[13] (Supplementary Fig. 1b). Increase in GFP signals was detected in mesophyll cells of *proATML1-nls-3xGFP* plants one day after removal of the leaf epidermis, which suggests that the *ATML1* promoter contains cis-regulatory sequences sufficient for the surgery-induced activation of *ATML1* expression in mesophyll cells (Fig. 1c, d, o). Previously, we have shown that a 101-bp sequence within the *ATML1* promoter is sufficient for the expression in the outermost cells of the embryos[13]. Therefore, we next examined whether the 101-bp sequence is sufficient also for the *ATML1* upregulation in mesophyll cells or not. We removed the leaf epidermis of *101×6-nls-3xGFP* plants, which produced the nuclear-localized triple GFP reporter under the control of six tandem repeats of

the 101-bp sequence[13] (Supplementary Fig. 1b). Increase in GFP signals was not detected one day after removal of the epidermis (4 of 4 independent lines; Fig. 1e, f), which suggests that regulatory sequences other than the 101-bp sequence are required for *ATML1* induction in mesophyll cells after surgery.

Next, to examine whether *ATML1* expression was increased only in the mesophyll cells located at the outermost position, we cleared *proATML1-nls-3xGFP* leaves with a clearing reagent, ClearSee[14]. *proATML1-nls-3xGFP* signals were detected in the outermost mesophyll cells but not in the inner mesophyll cells one day after removal of the epidermis (Fig. 1g–i and Supplementary Fig. 2a, b). These results suggest that *ATML1* transcription is activated specifically in the outermost mesophyll cells in a position-dependent manner.

To examine whether complete removal of the epidermis is required for *ATML1* activation in the mesophyll cells, we injured epidermal tissues. When epidermal cells of the first or second leaves in 4- or 9-day-old seedlings were damaged with a needle, an increase in *ATML1* promoter activity was detected in the subepidermal mesophyll cells beneath the damaged epidermis (10 of 12 and 9 of 10 leaves for 4- and 9-day-old seedlings, respectively; Fig. 1j–l, Supplementary Fig. 2c, d and Supplementary Fig. 3). These results suggest that the epidermal cell injury was sufficient to trigger *ATML1* induction in the underlying mesophyll cells.

### *ATML1* promoter activity was increased in mesophyll cells of a JA-insensitive mutant after removal of the epidermis

After removal or injury of the epidermis, the mesophyll cells should be exposed to increased environmental stress; exposure to air (e.g., oxidation) and dehydration were feasible candidates for the triggers of *ATML1* expression. However, *ATML1* upregulation was detected even when seedlings were grown in liquid medium after removal of the epidermis (see below), which suggests that oxidation and dehydration are not required for the *ATML1* induction.

Epidermis peeling might induce a wound response in underlying mesophyll cells, which could be a trigger of the *ATML1* induction. Jasmonate (JA) is known to mediate the wound response; *CORONATINE INSENSITIVE 1* (*COI1*), encoding an F-box protein, is a central component that promotes JA-mediated gene expression and is required for the wounding response[15,16]. *ATML1* upregulation was also detected in mesophyll cells of the *coi1* mutant, which suggests that *COI1*-mediated JA-signaling is not required for the *ATML1* induction in mesophyll cells although we cannot exclude the possibility that JA-independent wounding response might be involved in the *ATML1* induction (Supplementary Fig. 4).

### The *ATML1* induction in mesophyll cells after the injury was repressed by mechanical pressure

We noticed that mesophyll cells were often deformed and protruded from the leaf surface after epidermis removal (Fig. 2a and Supplementary Fig. 5). This observation implies that mesophyll cells are pressed by the epidermis and are released from the pressure after removal of the epidermis. Therefore, we examined whether the release from mechanical constraints is involved in the *ATML1* induction or not. To this end, we placed the epidermis-peeled *proATML1-nls-3xGFP* leaf between two coverslips and pressed it by using a paperclip (Fig. 2b). GFP positive cells were reduced in the pressed region compared with the non-pressed region (Fig. 2c, d, g). The cell length along the dorsoventral axis was not significantly different between the pressed outermost mesophyll cells and the intact L2 mesophyll cells overlaid with the epidermis (Supplementary Fig. 6). This observation suggests that our method can provide appropriate pressure to inhibit cell expansion of the outermost mesophyll cells. To examine the correlation between mesophyll cell expansion (i.e. pressure release) and *ATML1* induction, we focused on "boundary mesophyll cells" which are located near the boundary between peeled and unpeeled regions but

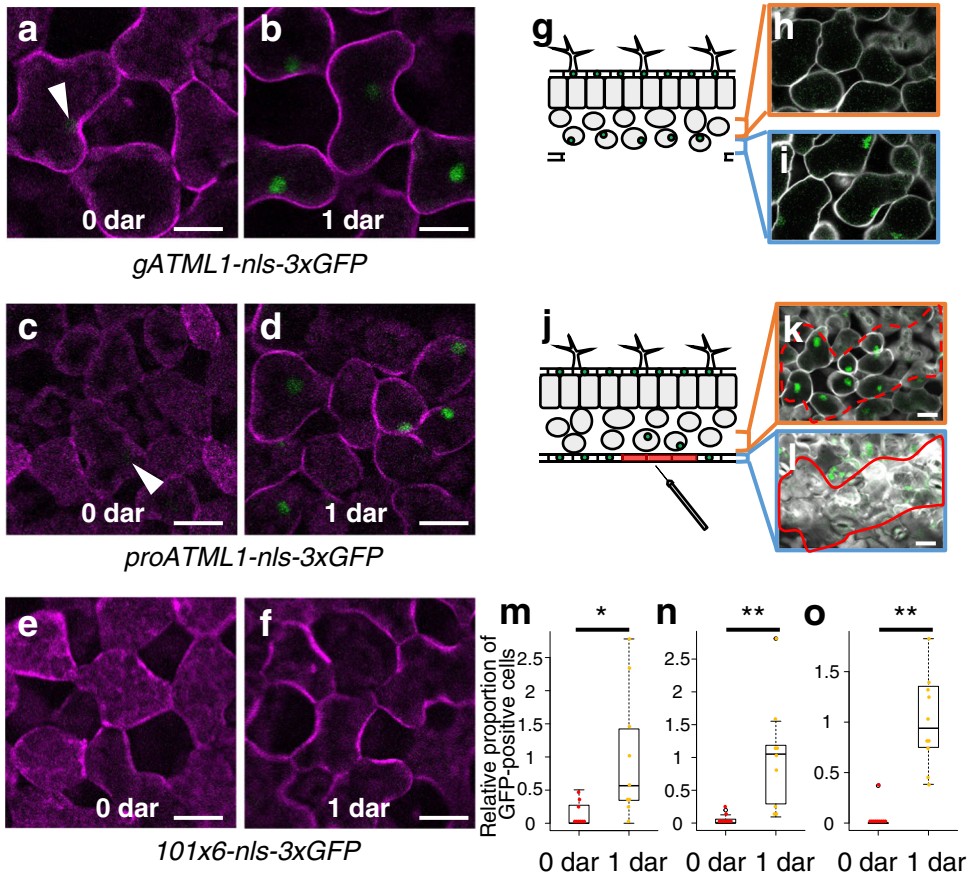

**Fig. 1 | *ATML1* transcription was activated in the outermost mesophyll cells after removal of the epidermis. a–f** The outermost mesophyll cells of *gATML1-nls-3xGFP* (**a, b**), *proATML1-nls-3xGFP* (**c, d**) and *101x6-nls-3xGFP* (**e, f**) leaves. The outermost mesophyll cells of 10-day-old plants were observed right after (**a,c,e**) and one day after (**b, d, f**) removal of the epidermis (0 dar and 1 dar, respectively; dar, day after removal of the epidermis). White arrowheads indicate *ATML1*-positive nuclei. **g** Schematic drawing of a transverse leaf section with a part of abaxial (lower) epidermis removed. **h, i** Inner mesophyll cells (**h**) and the outermost mesophyll cells (**i**) of a 10-day-old *proATML1-nls-3xGFP* seedling at 1 dar. **j** Schematic drawing of a leaf transverse section with a part of abaxial epidermis injured (highlighted in red). **k, l** The epidermis of a 9-day-old *proATML1-nls-3xGFP* leaf was damaged with a needle, and mesophyll (**k**) and epidermal (**l**) cells were observed one day after injury. The red dashed region in **k** mesophyll cells beneath the damaged epidermis; the red lined region in **l** the dead epidermis. Experiments in **a–f, h, i, k, l** were repeated three times with similar results. **m–o** The relative proportion of mesophyll cells showing GFP signals above the threshold was quantified in 10-day-old *gATML1-nls-3xGFP* (two independent lines; **m, n**) and *proATML1-nls-3xGFP* (**o**) seedlings at 0 dar and 1 dar. $n$ = nine biologically independent leaves except for 1 dar in **o** ($n$ = ten). Two-tailed Wilcoxon rank-sum test was used (*$P < 0.05$; **$P < 0.01$). In the box plots, the 25th percentile, the 50th percentile (central value) and the 75th percentile are marked by horizontal lines within the box. The ends of the whiskers indicate the maximum and minimum values within 1.5 x IQR from the box ends. Outliers are shown above the whiskers. Green, GFP; magenta, FM4-64; white, SR2200. Scale bars: 20 μm.

are still overlaid by the epidermis. Because these boundary mesophyll cells are expected to be under different levels of pressures from the partially damaged epidermal layer, we used these cells to examine the correlation between cell expansion and *ATML1* expression. Indeed, some boundary mesophyll cells showed invasion into the epidermal layer, suggesting a partial release from the pressure (Supplementary Fig. 7a, b). We found that most of those invaded cells showed *ATML1* expression, whereas few uninvaded mesophyll cells did (Supplementary Fig. 7c). These data suggest that there is a positive correlation between mechanical de-repression and *ATML1* induction. To test the possibility that decreased GFP signals were caused by the impaired viability of mesophyll cells after the pressure treatment, we utilized a transgenic line in which *GFP-ATML1* expression is constitutively induced by estradiol treatment[12] (*RPS5A»GFP-ATML1*). *RPS5A»GFP-ATML1* seedlings, with epidermis-peeled leaves pressed between coverslips, were grown in the liquid medium supplemented with estradiol. One day after incubation, GFP-ATML1 was induced in mesophyll cells in both pressed and non-pressed regions (12 of 12 plants; Supplementary Fig. 8a, b). To exclude the possibility that liquid medium may improve the cell viability, we also utilized the *R2D2* reporter, in which *mDII-nls-TdTomato* is expressed under the control of the constitutive *RPS5A*

promoter[17]. The TdTomato signal intensities in the exposed mesophyll cells were not significantly different between the non-pressed and pressed leaves grown on MS-agar plates (Supplementary Fig. 8c–e). These data suggest that the cell viability and general transcription/translation machinery are not impaired by the pressure treatment. Taken together, these results imply that the mechanical constraint by the epidermis is necessary for the repression of *ATML1* in mesophyll cells. Because *ATML1* expression in the differentiated epidermal cells was not affected by the pressure treatment (20 of 20 leaves; Supplementary Fig. 9), the mechanical pressure may efficiently repress only the de-novo expression of *ATML1*, as seen in the exposed mesophyll cell.

Mesophyll cell expansion induced by epidermis-peeling might be caused by high turgor pressure in these cells. To test whether mesophyll cell expansion or high turgor pressure is required for *ATML1* derepression, *proATML1-nls-3xGFP* plants were grown in liquid medium with low or high osmolality after removal of the epidermis. There were fewer GFP-positive mesophyll cells in high osmolality medium containing 400 mM mannitol than in low osmolality medium with 80 mM mannitol (Fig. 2e, f, h). We also performed the same experiments using the *RPS5A»GFP-ATML1* line and found that *GFP-ATML1* was

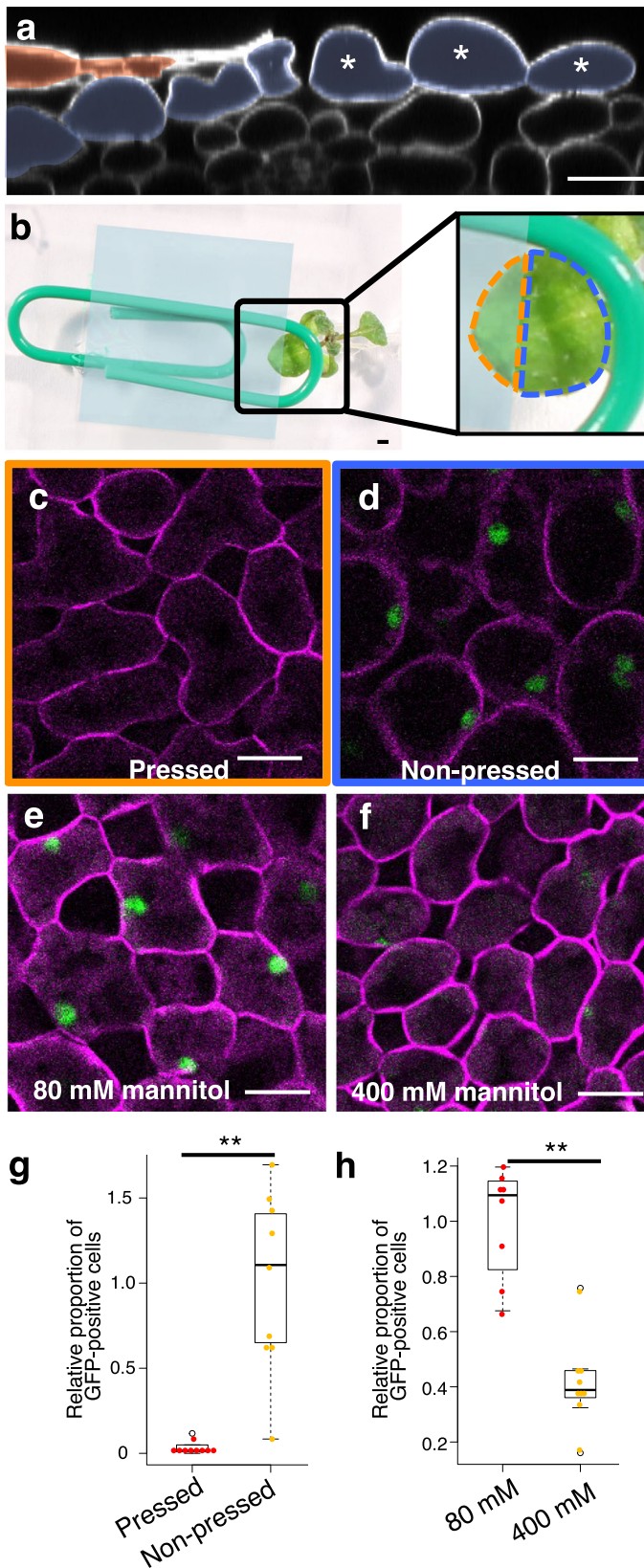

**Fig. 2 | *ATML1* induction in mesophyll cells was repressed by mechanical pressure and under hyperosmotic conditions. a** Optical transverse section of a 10-day-old *proATML1-nls-3xGFP* leaf at 1 dar. Red, epidermal cells; blue, mesophyll cells; asterisks, outermost mesophyll cells. **b** After epidermis peeling, the first or second leaf of a 9-day-old *proATML1-nls-3xGFP* seedling was placed between two coverslips and pressed with a paperclip. The orange dashed region, pressed region; the blue dashed region, non-pressed region. **c, d** *proATML1-nls-3xGFP* signals in the outermost mesophyll cells in the pressed (**c**) and non-pressed (**d**) regions at 1 dar. **e, f** *proATML1-nls-3xGFP* signals in the outermost mesophyll cells of 10-day-old seedlings grown in 80 mM (**e**) and 400 mM (**f**) mannitol-containing liquid MS medium for 24 h after surgery. Experiments in **a** and **c-f** were repeated three times with similar results. **g, h** The relative proportion of mesophyll cells showing GFP signals above the threshold was quantified in the pressed and non-pressed regions (**g**) and in 80 mM and 400 mM mannitol-treated seedlings (**h**). *n* = nine biologically independent leaves except for 80 mM in **h** (*n* = eight). Two-tailed Wilcoxon rank-sum test was used (**$P < 0.01$). In the box plots, the 25th percentile, the 50th percentile (central value) and the 75th percentile are marked by horizontal lines within the box. The ends of the whiskers indicate the maximum and minimum values within 1.5 x IQR from the box ends. Outliers are shown above and below the whiskers. Green, GFP; magenta; FM4-64. Scale bars: 20 μm in **a**, **c-f**; 1 mm in **b**.

treatment significantly decreased cell expansion, as well as, *ATML1* expression of exposed mesophyll cells, inhibition of cell expansion or plasma membrane stretching by the mechanical pressure from the epidermis may mediate *ATML1* repression in the inner mesophyll cells. Taken together, these results are consistent with the conclusion that the derepression of *ATML1* transcription in mesophyll cells after the injury depends on their mechanical environments.

### Increase in cortical microtubule formation in mesophyll cells was necessary for the derepression of *ATML1* transcription

Because cortical microtubules are aligned along the maximum mechanical stress directions in the shoot apical meristems, changes in mechanical constraints of the mesophyll cells might also affect microtubule organization[18]. Therefore, we next observed cortical microtubules by using the tubulin marker *35S-GFP-TUA6*[19]. In the intact leaves of 6-day-old seedlings, cortical microtubules were detected as foci at the anticlinal cell boundaries of the epidermis but not of sub-epidermal cells (16 of 17 leaves; Supplementary Fig. 10a,b). These observations suggest that mesophyll cells have less cortical micro-tubules compared with the epidermis. Consistently, only a few cortical microtubule arrays were detected in mesophyll cells right after the removal of the epidermis. However, the density of cortical micro-tubules in mesophyll cells was increased three hours after epidermis peeling (Fig. 3a–c). The cortical microtubule arrays were still promi-nent in the outermost mesophyll cells one day after the removal of the epidermis (21 of 26 leaves; Supplementary Fig. 10c,d). In summary, in regard to the cortical microtubule density, the exposed mesophyll cells resembled the intact epidermal cells more than the inner meso-phyll cells of untreated younger leaves.

　　Next, we tested whether or not *ATML1* derepression is detected when the cortical microtubule formation is disrupted. Treatment with oryzalin, which disrupts microtubules, severely reduced the density of microtubules in exposed mesophyll cells (Fig. 3d–f). *proATML1-nls-3xGFP*-positive mesophyll cells were also decreased in seedlings trea-ted with oryzalin for 24 h after surgery (Fig. 3g–i). These observations suggest that the increase of cortical microtubule formation is required to derepress *ATML1* promoter activity in mesophyll cells. We also examined whether cortical microtubule formation in mesophyll cells is affected by the mechanical pressure. We pressed leaves of *35S-GFP-TUA6* plants after the removal of the epidermis and observed cortical microtubules in mesophyll cells. The density of cortical microtubules was similarly increased in both the pressed and non-pressed regions three hours after epidermis removal (Fig. 3j–l). This observation sug-gests that the pressure treatment does not affect cortical microtubule

induced in the outermost mesophyll cells upon estradiol treatment, regardless of mannitol concentration (Supplementary Fig. 8f, g). This result suggests that the mannitol treatment did not impair the cell viability. The high osmolality of the medium, which absorbs water from the cells and inhibits cell expansion, essentially mimicked the presence of overlying epidermal cells. Considering that the pressure

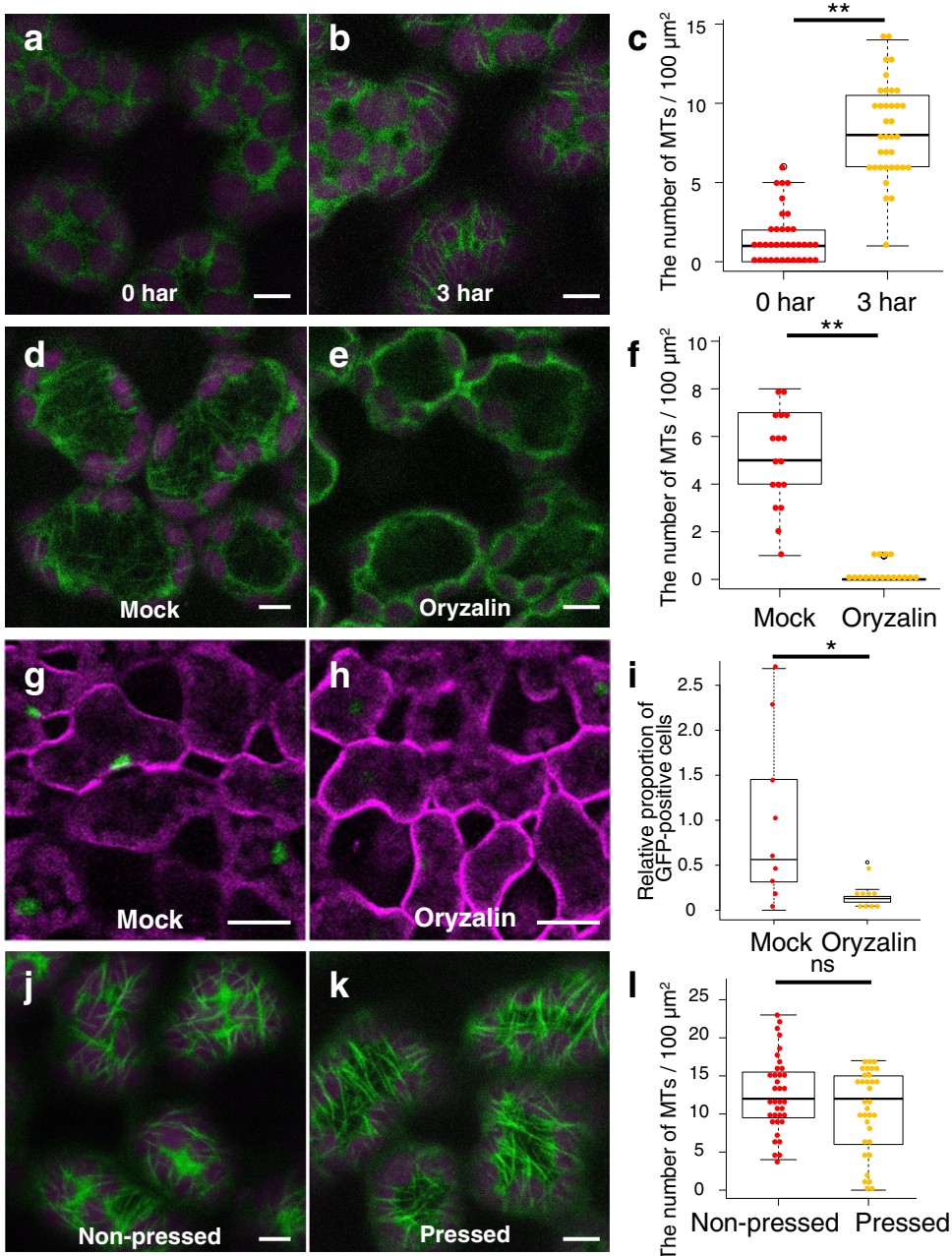

**Fig. 3 | Cortical microtubule formation was required for *ATML1* derepression in the outermost mesophyll. a**, **b** *35S-GFP-TUA6* signals in mesophyll cells of 9-day-old seedlings were observed right after (**a**) and three hours after (**b**) removal of the epidermis (0 har and 3 har, respectively; har, hour after removal of the epidermis). **c** Quantification of the cortical microtubule (MT) density in mesophyll cells of 9-day-old *35S-GFP-TUA6* seedlings at 0 har and 3 har. *n* = 36 cells from 12 biologically independent leaves (three cells from each leaf). Two-tailed Welch's *t*-test was used (**P* < 0.01). **d**, **e** GFP-TUA6 protein localization in mesophyll cells of 9-day-old seedlings grown in liquid MS medium supplemented with 0.1% DMSO (**d**; Mock) or 30 μM oryzalin (**e**; Oryzalin) for three hours after epidermis removal. **f** The cortical microtubule density in the outermost mesophyll cells was quantified in mock and oryzalin-treated *35S-GFP-TUA6* leaves at three har. *n* = 18 cells from six biologically independent leaves (three cells from each leaf). Two-tailed Welch's *t*-test was used (**P* < 0.01). **g**, **h** Mesophyll cells of the epidermis-peeled first or second leaves in 10-day-old *proATML1-nls-3xGFP* seedlings grown in liquid MS medium supplemented with 0.1% DMSO (**g**) or 30 μM oryzalin (**h**) for 24 h after surgery. **i**, The relative proportion of mesophyll cells showing *proATML1-nls-3xGFP* signals above the

threshold was quantified in mock and oryzalin-treated plants. *n* = nine biologically independent leaves. Two-tailed Wilcoxon rank-sum test was used (**P* < 0.05). **j**, **k** GFP-TUA6 protein localization in mesophyll cells of 9-day-old seedlings grown without (**j**; Non-pressed) or with (**k**; Pressed) pressure treatment on peeled leaves for three hours after surgery. Each experiment was repeated three times for **a**, **b**, **g**, **h** and twice for **d**, **e**, **j**, **k** with similar results. **l** The cortical microtubule density was quantified in the outermost mesophyll cells grown for 3 h without or with pressure treatment after surgery. n = 36 cells from 12 biologically independent leaves for non-pressed leaves and *n* = 33 cells from 11 biologically independent leaves for pressed leaves. Three cells from each leaf were examined. Two-tailed Welch's *t*-test was used (ns, *P* ≥ 0.05). For all the box plots, the 25th percentile, the 50th percentile (central value) and the 75th percentile are marked by horizontal lines within the box. The ends of the whiskers indicate the maximum and minimum values within 1.5 x IQR from the box ends. Outliers are shown above the whiskers. Green, GFP in **a**, **b**, **d**, **e**, **g**, **h**, **j**, **k**; magenta, chlorophyll autofluorescence in **a**, **b**, **d**, **e**, **j**, **k** and FM4-64 in **g**, **h**. Scale bars: 5 μm in **a**, **b**, **d**, **e**, **j**, **k** and 20 μm in **g**, **h**.

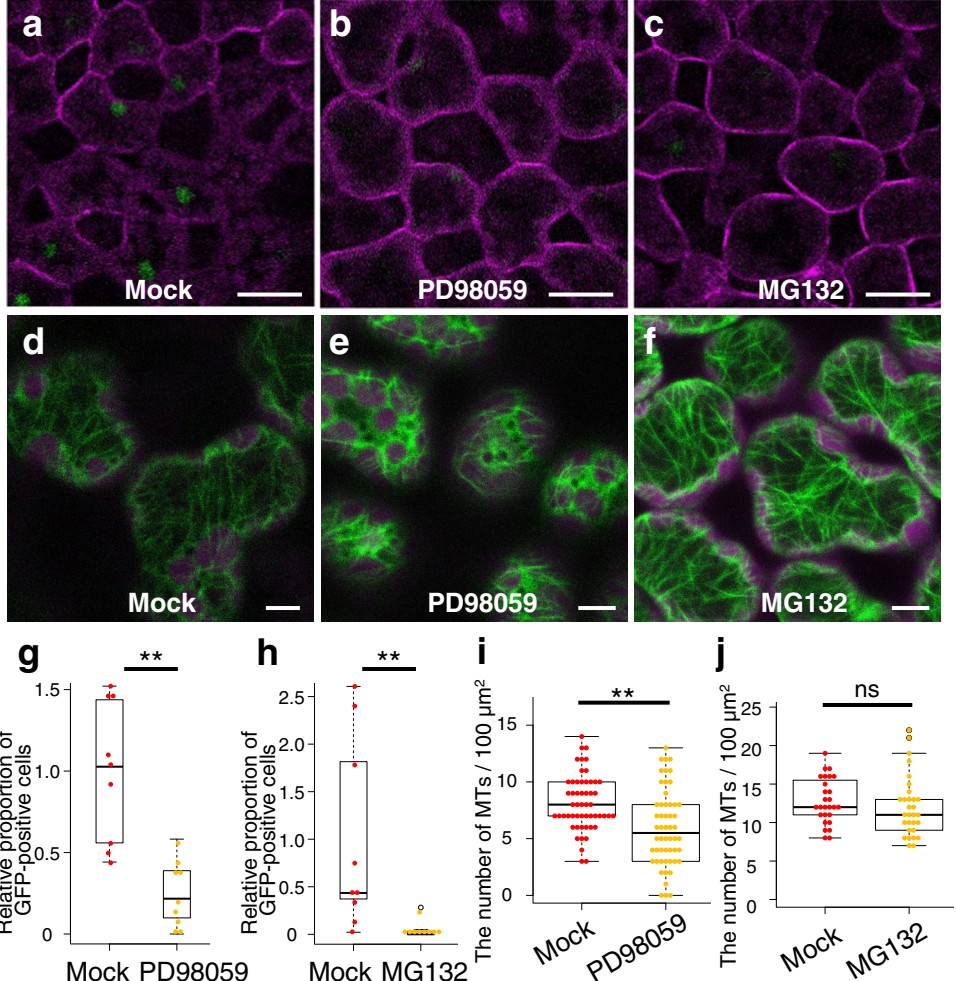

**Fig. 4 | MAPK signaling and proteasome activities were required for *ATML1* derepression in the outermost mesophyll cells. a–c** Mesophyll cells of the epidermis-peeled first or second leaves in 10-day-old *proATML1-nls-3xGFP* seedlings grown in liquid MS medium supplemented with 0.1% DMSO (**a**, Mock), 25 μM MAPKK inhibitor PD98059 (**b**, PD98059) or 10 μM proteasome inhibitor MG132 (**c**, MG132) for 24 h after surgery. **d–f** GFP-TUA6 protein localization in the outermost mesophyll cells of 9-day-old seedlings grown in liquid MS medium supplemented with 0.1% DMSO (**d**), 25 μM PD98059 (**e**) or 10 μM MG132 (**f**) for three hours after surgery. Each experiment was repeated three times for **a–e** and twice for **f** with similar results. **g**, **h** The relative proportion of the outermost mesophyll cells showing *proATML1-nls-3xGFP* signals above the threshold was quantified in mock, PD98059 or MG132-treated seedlings at 1 dar. *n* = nine biologically independent leaves. Two-tailed Wilcoxon rank-sum test was used (**\*\*P* < 0.01). **i**, **j** The cortical microtubule density in the outermost mesophyll cells was quantified in mock, PD98059 or MG132-treated *3SS-GFP-TUA6* leaves at 3 har. *n* = 54 cells from 18 biologically independent leaves for **i**, n = 27 cells from nine biologically independent leaves for Mock in **j** and *n* = 30 cells from ten biologically independent leaves for MG132 in **j**. Three cells from each leaf were examined. Two-tailed Welch's t-test was used (**\*\*P* < 0.01; ns, *P* ≥ 0.05). For all the box plots, the 25th percentile, the 50th percentile (central value) and the 75th percentile are marked by horizontal lines within the box. The ends of the whiskers indicate the maximum and minimum values within 1.5 x IQR from the box ends. Outliers are shown above the whiskers. Green, GFP in **a–f**; magenta, FM4-64 in **a–c** and chlorophyll autofluorescence in **d–f**. Scale bars: 20 μm in **a–c** and 5 μm in **d–f**.

formation in mesophyll cells. In addition, the increased cortical microtubules in the pressed regions, which showed decreased *ATML1* induction, indicate that cortical microtubule formation is necessary but not sufficient for *ATML1* derepression in the mesophyll cells.

### MAPK cascade and protein degradation might be necessary for the *ATML1* derepression

To examine signaling pathways involved in the de-novo *ATML1* induction, we looked for other inhibitors that reduce the induction of *proATML1-nls-3xGFP* signals in mesophyll cells. We identified two inhibitors, PD98059 and MG132, that each decreased the proportion of *proATML1-nls-3xGFP* positive mesophyll cells after epidermis removal (Fig. 4a–c, g, h). PD98059 is known as a MAPKK inhibitor[20,21] and MG132 is a proteasome inhibitor[22,23]. This finding therefore implies that MAPK signaling and proteasome-mediated protein degradation are necessary for the *ATML1* derepression in the mesophyll cells. We also

examined whether these inhibitors affect the formation of cortical microtubules in mesophyll cells. The PD98059 treatment only slightly reduced the density of cortical microtubules and the MG132 treatment did not affect the formation of cortical microtubules in the outermost mesophyll cells after removal of the epidermis (Fig. 4d–f, i, j). These observations suggest that MAPK signaling and proteasome activities are not essential for the cortical microtubule formation in these cells.

Oryzalin, PD98059 and MG132 did not reduce *proATML1-nls-3xGFP* signals in the leaf epidermis or embryos (Supplementary Fig. 11). *ATML1* expression is initiated in the one-cell stage embryo and is stably maintained by a positive feedback through the L1 box, an ATML1 binding site, in the normal developmental conditions[13,24]. Therefore, target components of oryzalin, PD98059 and MG132 may be required for de-novo expression of *ATML1* during mesophyll cell re-specification but not for the already established expression of *ATML1* in the outermost cells.

## Outer tissue injury caused epidermis regeneration from the inner-lineage tissue of the leaf

The derepression of *ATML1* in the outermost mesophyll cells might result in epidermal cell differentiation in these cells. Therefore, we investigated whether the *ATML1*-positive mesophyll cells can change their fate and regenerate the epidermis after removal or injury of the overlying epidermis. In the original experimental conditions, we did not find an epidermal-related trait in the outermost mesophyll cells even when plants were grown for seven days after removal of the epidermis (Supplementary Fig. 12a). This might be because ATML1 protein accumulation was inhibited in the inner-lineage cells as demonstrated previously in the embryos[12]. Therefore, we checked ATML1 protein accumulation in mesophyll cells using the *gATML1-3xGFP* reporter, in which the *3xGFP-ATML1* fusion gene was expressed under the native *ATML1* regulatory sequence[12] (Supplementary Fig. 1b). To easily spot the location of the mesophyll cells beneath the damaged epidermal cells, we injured trichomes on the first or second leaves of 9-day-old *gATML1-3xGFP* seedlings (Fig. 5a). One day after the surgery, ATML1 protein was not detected in mesophyll cells beneath the damaged trichomes (39 of 39 trichomes; Supplementary Fig. 12b, c). This observation might suggest that mesophyll cells used in our experiments were fully differentiated and did not have a potential to change their cell fates. To test this possibility, we next utilized younger seedlings to examine ATML1 protein accumulation. ATML1 protein was detected in the nuclei of mesophyll cells beneath the killed trichomes of 6-day-old *gATML1-3xGFP* seedlings (20 of 45 trichomes; Fig. 5b). This observation implies that mesophyll cells of younger (i.e. 6-day-old) seedlings have a weaker inhibitory effect on ATML1 protein accumulation and could be re-specified into epidermal cells. To observe cell identity change in mesophyll cells after epidermal cell death, we damaged, with a needle, wider regions of the abaxial surface of the first or second leaves in 6-day-old *proATML1-nls-3xGFP* seedlings. Accumulation of propidium iodide (PI), a dead cell stain, in the nuclei and the absence of *proATML1-nls-3xGFP* expression right after the injury confirmed the epidermal cell death in the damaged region (Supplementary Fig. 13a). Five days after the injury, most of the leaf surface cells expressed *proATML1-nls-3xGFP*, suggesting the regeneration of epidermal cells. We noticed that dents were formed on the leaf surface, presumably in the damaged region (Fig. 5c, d). In the dent region, *ATML1*-positive cells with small lobes, which are characteristics of young pavement cells, were observed (19 of 22 leaves; Supplementary Fig. 13b). These *ATML1*-positive cells showed plastid autofluorescence similar to that seen in the intact pavement cells (Supplementary Fig. 13c–e). Stomatal guard cell formation was also observed in the damaged region (17 of 25 leaves; Supplementary Fig. 13b). In order to know whether trichome cells can be also regenerated after epidermis injury, we damaged the trichome-rich adaxial leaf surface. In the damaged region we found trichome-like cells that expressed the trichome marker *GLABRA2 (GL2)* but were less branched and not surrounded by typical accessory cells, suggesting that they were generated from an unusual developmental process (Supplementary Fig. 14a)[25]. Taken together, these observations suggest that three epidermal cell types, pavement cells, guard cells and trichomes, were regenerated after epidermal cell injury.

We noticed that stomata were formed in the inner tissues below the damaged region after abaxial tissue injury, as revealed by 3D reconstruction images (6 of 29 leaves; Fig. 5c, d). Those inner guard cells and their surrounding cells expressed the *proATML1-nls-3xGFP* marker (Fig. 5c). The inner guard cell formation was also induced when the adaxial outer tissues of the first or second leaves were damaged, as evidenced by *KAT1-GUS* guard cell marker expression (Supplementary Fig. 14b)[26]. We then looked for other epidermal cell types formed in the inner tissues after epidermal injury. Intact pavement cells show dot-like small plastid autofluorescence whereas mesophyll cells and guard cells show stronger and larger autofluorescence signals from chloroplasts (Supplementary Fig. 13c, d). In the damaged leaves, we found *ATML1*-positive inner cells that showed small plastid auto-fluorescence characteristic of the pavement cells (36 of 216 cells; Supplementary Fig. 14c). The epidermal cell regeneration in the inner tissues supports the ideas that inner mesophyll cells were re-specified-into-epidermal-cells when the outer tissues of the leaves were damaged.

Older mesophyll cells, in which ATML1 protein accumulation is blocked, could not be re-specified as epidermis, suggesting a role of *ATML1* in promoting epidermal cell regeneration from mesophyll cells. To further evaluate the necessity of *ATML1* for the re-specification of mesophyll cells into the epidermis, we used an *atml1* null mutant allele (*atml1-3*)[27]. The outer tissues of the first and second leaves of 6-day-old *atml1-3* seedlings were killed with a needle, which was confirmed by PI staining (Supplementary Fig. 15a). After five days, we found that the outermost cells of *atml1-3* in the damaged region often protruded and were morphologically different from the epidermis (Supplementary Fig. 15b, c). Plastids in these protruded cells were larger than those of the intact pavement cells (Supplementary Fig. 15b, d), suggesting a failure in epidermal cell regeneration. In addition to the *atml1-3* mutant, we also utilized the *RPS5A»ATML1-SRDX* line, in which expression of *ATML1* downstream genes is repressed upon the estra-diol treatment[28]. *ATML1-SRDX* was induced just after the first or second leaves of 6-day-old seedlings were damaged and the treated seedlings were grown for another five days. In *ATML1-SRDX* induced leaves, cells in the damaged region protruded from the surface and formed callus-like tissues (22 of 32 leaves), which was not observed in control leaves (0 of 46 leaves). To evaluate the efficiency of epidermal regeneration, we used the guard cell formation in the inner tissues as an indicator of mesophyll cell re-specification. This method helped us to distinguish regenerated epidermal cells from the original epidermis. While injury-induced inner guard cell formation was found in DMSO-treated control leaves (11 of 46 leaves), it was not detected in *ATML1-SRDX* induced leaves (0 of 32 leaves; Supplementary Fig. 15e). Taken together, these results suggest that activation of *ATML1* and/or its target genes is required for the regeneration of the epidermal cells from the inner lineage cells.

In summary, we conclude that position- and age-dependent transcriptional/post-transcriptional regulations of *ATML1* determine the potential for epidermal cell regeneration in plants.

## Discussion

Forty-five years ago, sector analyses in periclinal chimeric albino leaves suggested that L1-derived cells displaced to the L2 cell layer develop into mesophyll cells[7]. However, it has not been tested whether or not mesophyll-derived cells can acquire an epidermal cell fate when displaced to the outermost positions. The epidermal cell regeneration from the inner-lineage cells after epidermal cell injury in the present study strongly supports the idea that the "cell position" is important to determine epidermal or mesophyll cell identity. We propose that *ATML1* derepression is one of the molecular mechanisms that explain this cell fate change. First, de-novo transcriptional derepression of the master gene *ATML1* preceded the epidermal fate acquisition in inner-lineage cells. Second, young mesophyll cells that can accumulate ATML1 protein in the nuclei are re-specified as epidermis. Third, nuclear accumulation of ATML1 is decreased in the epidermis-derived cells displaced to the inner cell layer after periclinal cell division[12]. Lastly, repression of *ATML1* and/or its target genes impaired epidermal cell regeneration from the mesophyll cells. We found that cortical microtubules, MAP kinases, and proteasomes are candidate components of signaling pathways that induce *ATML1* transcription in the outermost mesophyll cells. Moreover, our results imply that the mechanical pressure from the overlying epidermis is involved in the repression of *ATML1* in the mesophyll cells. Mechanical environments are different between the outermost and inner tissues; the outermost

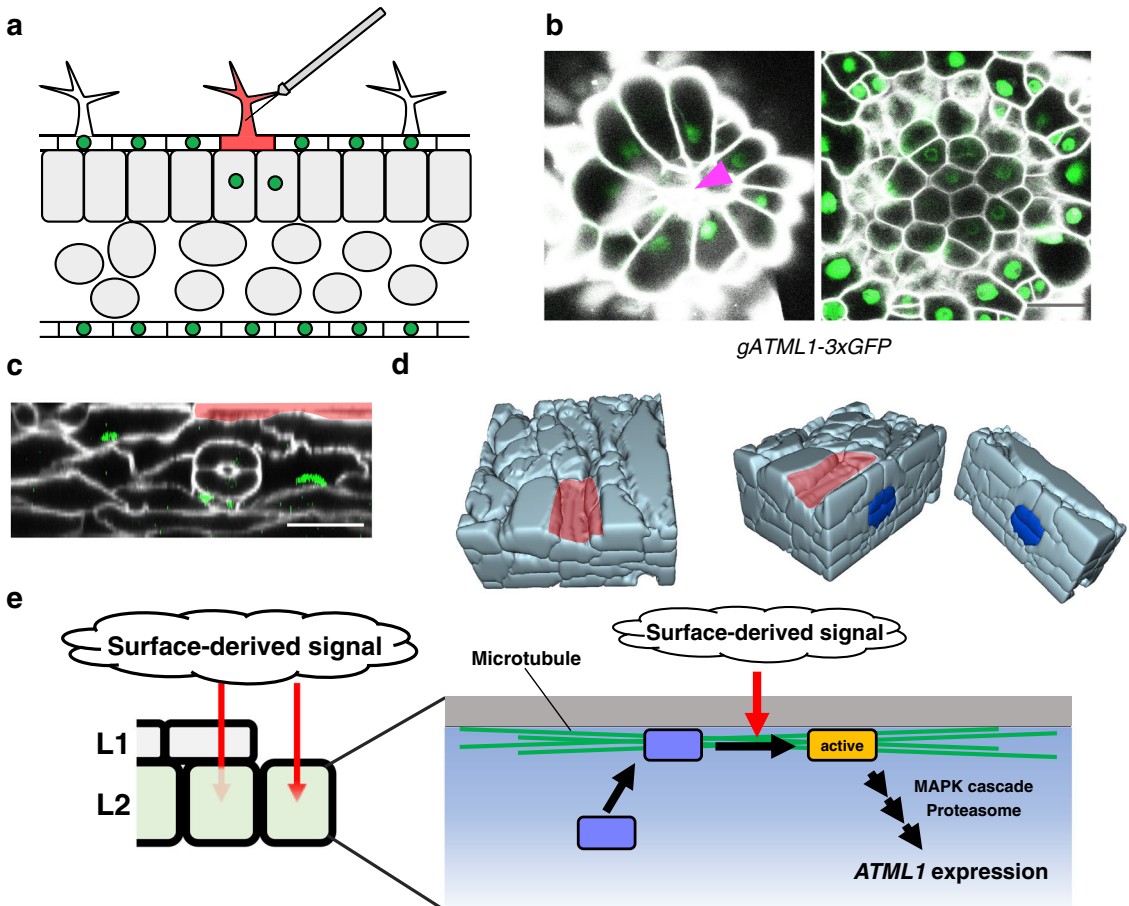

*gATML1-3xGFP*

**Fig. 5 | The epidermis was regenerated from mesophyll cells after outer tissue injury. a** Schematic drawing of a transverse leaf section showing trichome injury. **b** ATML1 protein accumulation in mesophyll cells beneath the killed trichomes. After trichomes of the first or second leaves in 6-day-old *gATML1-3xGFP* seedlings were damaged with a needle, the seedlings were grown for 24 h. Left, leaf epidermis with a killed trichome (arrowhead); right, mesophyll cells beneath the epidermis of the same leaf as in the left panel. **c, d** Stomata formation from the inner lineage tissues of the leaves after outer tissue injury. After the outer tissues of leaves in 6-day-old *proATML1-nls-3xGFP* seedlings were damaged with a needle, the seedlings were grown for five days. **c** shows an optical transverse section of the leaf. Note that inner guard cells and their surrounding cells expressed the *proATML1-nls-3xGFP* marker. In **d** the mesophyll-derived inner stoma in **c** is shown in 3D models reconstructed from optical sections. The left model is cut in half to generate middle and right models. Experiments in **b**, **c** were repeated three times with similar results. **e** Schematic model for molecular mechanisms underlying the derepression of *ATML1* in the outermost cells. It has been suggested that the epidermis is under lateral tension[29]. After removal of the epidermis, the outermost mesophyll cells are expanded vertically, which suggests that plasma membrane of the outermost mesophyll cells is under the same mechanical environment as the epidermal cells (i.e., stretched laterally). Recent studies suggest that stretching of the plasma membrane promotes the activity of certain plasma membrane channels[30]. Therefore, we speculate that lateral tension to the outermost cells might stretch plasma membrane and activate unknown plasma membrane factors and their downstream signaling pathways that lead to derepression of *ATML1*. In this model, cortical microtubules formed after cell exposure to the organ surface may act as scaffolds to recruit unknown factors (blue rectangles) that promote epidermal cell specification. Those epidermal fate-promoting factors are activated beneath the plasma membrane by plasma membrane stretch and turn on the downstream signaling that promotes *ATML1* transcription. MAPK signaling and the proteasome might be required to activate those plasma membrane factors or downstream factors. Blue, inner guard cells; the red region, presumably damaged epidermal tissues. Green, GFP in **b**, **c**; white, PI in **b** and SR2200 in **c**. Scale bars: 20 μm.

tissue is stretched and the inner tissue is compressed, which may be used as positional cues[29]. We assume that the mechanical pressure exerted by the epidermis inhibits cell expansion as well as *ATML1* upregulation in the mesophyll cells. First, mesophyll cells expanded outward when the overlying epidermis was removed, suggesting that mesophyll cells were pressed by mechanical pressures exerted by the epidermis. Second, mesophyll cell expansion after epidermal cell removal was positively correlated with *ATML1* induction. Third, when the cell expansion of exposed mesophyll cells was prevented by the mannitol treatment or pressure treatment with a paper clip, *ATML1* upregulation was repressed in those mesophyll cells. These results are taken as evidence that mesophyll cell expansion, triggered by the removal of the epidermis, is at least one of the causes of de-novo *ATML1* expression during regeneration. However, we cannot exclude the possibility that other signals caused by the damaged epidermis

(e.g., wounding response) are also involved in *ATML1* de-repression. Moreover, our results are based on observations during epidermis regeneration. Therefore, epidermal cell specification or maintenance during normal development may use different molecular mechanisms. We made a working model for de-novo *ATML1* induction in the mesophyll cells by integrating our findings and ideas (Fig. 5e). In this model, we propose that cortical microtubules recruit unknown factors near the plasma membrane. The lateral stretch in the outermost cells activates those factors beneath the plasma membrane, leading to the derepression of *ATML1*. The MAPK pathway and proteasome activity are required for this process (Fig. 5e). This study not only revealed molecular targets for epidermal/mesophyll cell fate change in the leaves – a long-standing question – but also provides new mechanistic insights and a powerful tool to investigate position-dependent cell fate decision in plants.

## Methods

### Plants materials and growth conditions

*proATML1-nls-3xGFP*, *101×6-nls-3xGFP*, *gATML1-nls-3xGFP*, *RPS5A»GFP-ATML1*, *RPS5A»ATML1-SRDX*, *R2D2*, *KAT1-GUS*, *GL2-GUS*, *atml1-3* and *35S-GFP-TUA6* plants have been described previously[10,12,13,17,19,25–28]. *coi1* (SALK_035548) was obtained from the ABRC. Seeds were sown on Murashige and Skoog (MS) plates supplemented with 1% sucrose and solidified with either 0.4% phytagel or 1% agar. After incubation at 4 °C for more than two days, plates were moved to the growth chamber at 22 °C. The day when plates were transferred to the growth chamber is defined as day 0. After two or three weeks, seedlings were moved to soil and grown at 18 or 22 °C. Plants in Supplementary Fig. 3, 6, 8c, d, f, g, 9, 10, 13, 14 and 15 were grown on MS-agar plates under 16 h light condition and otherwise grown on MS-phytagel plates under continuous light.

### Surgical epidermis removal and injury

For epidermis removal, the first or second leaves of 9-day-old seedlings were used. The abaxial epidermis of the rosette leaf was removed by the following method. The outer tissue of the petiole was cut transversely with an injection needle under a dissection microscope. The cut edge was picked up with forceps and pulled toward the tip of the leaf to peel the epidermis. For the pressure treatment, peeled or intact leaves were placed between two cover slips and pressed by using a paper clip (28 mm in length, Kohnan). After the surgical treatment, seedlings were grown on MS plates or in liquid MS medium in the growth chamber at 22 °C. For the *coi1* mutant, we utilized F3 generation seeds collected from plants heterozygous for the *coi1* mutation and homozygous for *proATML1-nls-3xGFP*. F3 plants without carrying the *coi1* mutation were used as a control (WT).

For epidermis injury, the first or second leaves of 4-, 6- or 9-day-old seedlings were used. The abaxial or adaxial surface of the leaves was gently scratched with a needle (Roboz, RS-6064) under a dissection microscope to damage the outer tissues of the leaves. For the detection of ATML1 protein accumulation, trichomes of the leaves were pricked with a needle under a dissection microscope. After damaging the epidermis, the seedlings were grown on MS plates for indicated days in the growth chamber at 22 °C under constant light or 16 h light condition. *RPS5A»ATML1-SRDX* seedlings were grown on 0.0005% dimethyl sulfoxide (DMSO) or 0.1 μM estradiol containing MS-agar plates for five days after damaging. For the dead cell detection before regeneration, seedlings were mounted in 10 μg/ml PI solution right after injury and subjected to confocal laser scanning microscopy. After microscopy, the PI-stained seedlings were washed with water and grown on MS plates for five days to observe regeneration.

### Chemical treatment

To treat plants with chemicals after removal of the epidermis, seedlings were grown in 80 mM mannitol-containing liquid-MS medium supplemented with chemicals (30 μM oryzalin, 25 μM PD98059, 10 μM MG132, or 10 μM estradiol). All chemicals were dissolved in dimethyl sulfoxide (DMSO) as stock solutions. As a control, the same amount of DMSO was added to liquid-MS medium. Liquid-MS medium containing 400 mM mannitol was used for the hyperosmotic condition. Seedlings were grown in the growth chamber at 22 °C. Ovules were cultured in liquid Nitsch medium (5% trehalose), supplemented with chemicals or DMSO, at 22 °C for indicated days[12].

### Confocal laser scanning microscopy and image analysis

LSM710 (Carl Zeiss), FV1000 (Olympus) and Leica Stellaris (Leica) confocal laser scanning microscopes were used to detect GFP, TdTomato, GUS, plastid autofluorescence, FM4-64, PI and SCRI Renaissance 2200 (SR2200) signals. GUS signals were detected by using the reflection mode on Leica Stellaris. Mesophyll cells after removal of the epidermis were stained with 20 μM FM4-64 (Sigma Aldrich) in MilliQ

water or 1 μl/ml SR2200 (Renaissance Chemicals) in phosphate buffered saline (PBS) for more than five minutes. To observe plastid autofluorescence after damaging the outer tissues, leaves were fixed in 4% paraformaldefyde in PBS containing 2 μl/ml SR2200 for more than one hour at room temperature. Leaves were washed twice with PBS before observation. After damaging trichomes, leaves were mounted in 100 μg/ml PI (Nacalai) solution in MilliQ water for the staining of cell walls and dead cells. To observe the inner tissues of leaves, histological sections were made for Supplementary Figs. 6 and 14c, otherwise the ClearSee method was used[14]. For the histological sectioning, *proATML1-nls-3xGFP* leaves were fixed in 4% paraformaldehyde in PBS for one hour and washed once with PBS. The leaves were embedded in 4% agarose in PBS and cross sections were made with a vibratome. The cross sections were stained in 1 μl/ml SR2200 in ClearSee or PBS. For the ClearSee method, leaves were submerged in 4% paraformaldehyde in PBS containing 2 μl/ml SR2200 and were placed at room temperature for more than 30 min. The leaves were washed twice with PBS and cleared with the ClearSee solution.

The levels of *ATML1* induction and *mDII-TdTomato* expression were quantified by using Fiji (fiji.sc). To calculate the proportion of GFP-positive cells, z-stack images of the mesophyll cells were obtained. Maximum projection images of the GFP channel were generated from the z-stack series and were used to count the number of nuclei showing GFP signals above the threshold. Maximum projection images of the FM4-64 channel were used to count the total number of the outermost mesophyll cells. The proportion of GFP-positive cells was determined by calculating the number of GFP-positive mesophyll cells per total number of mesophyll cells in the observed field. For calculating the proportion, experiments were repeated three times; two to four leaves were used in each experiment. In total, nine leaves were used for quantification in Figs. 1m, n, 2g, 3i, 4g, h, and Supplementary Fig. 4c. Nine and ten leaves were used in experiments at 0 dar and 1 dar in Fig. 1o, respectively. Eight and nine leaves were used for quantification in 80 mM and 400 mM mannitol-treated plants, respectively. The proportions were normalized within each experiment by the average of the proportions in 1 dar samples for Fig. 1m–o, nonpressed samples for Fig. 2g, 80 mM treated samples for Fig. 2h, mock treated samples for Figs. 3i, 4g, h, and *coi1* samples for Supplementary Fig. 4c. For *mDII-TdTomato* quantification, the focal plane in which the TdTomato signal was the brightest was selected for each nucleus from the z-stack series and used for the measurement of signal intensity. TdTomato signal intensities in randomly selected three to five nuclei were averaged for each leaf. Fifteen non-pressed leaves and nine pressed leaves were used for the quantification.

For cell length measurement, 10 independent leaves were used. Cell lengths of 10 outermost mesophyll cells or 5 mesophyll cells beneath the epidermal cells in each leaf were averaged for Supplementary Fig. 5. For Supplementary Fig. 6, cell lengths of five cells in each category were averaged in each leaf. Data analysis was performed using R (https://www.R-project.org/). Transverse optical sections and 3D reconstruction were made by using Fiji.

To quantify the correlation between cell invasion and *ATML1* expression in the boundary mesophyll cells, we used z-stack images of mesophyll cells, which are located near the boundary between peeled/unpeeled regions and are overlaid by the epidermis, in the first or second leaf of 10-day-old *proATML1-nls-3xGFP* seedlings at 1 dar. We made optical cross-sections to examine whether boundary mesophyll cells are invaded toward the epidermis. When the outermost ends of mesophyll cells reached the middle depth of the overlaying epidermal layer, the mesophyll cells were judged as "invaded". Seventy-eight cells from eight leaves were used for the quantification.

The density of protruded cells in the damaged region was calculated by dividing the number of protruded cells by the damaged

area. We defined the protruded cell as the surface cell that has intercellular gap spaces between adjacent surface cells and the damaged region as the area where the protruded cells and less lobed pavement cell-like cells were detected. For the quantification of *proATML1-nls-3xGFP* and *atml1-3* regeneration, 20 and 41 leaves were used, respectively.

The sizes of the plastids were measured at the focal planes in which the autofluorescence was the largest for each plastid. Three or four plastids were randomly selected in each cell; plastid sizes in three cells of each category were measured and averaged in each leaf. Eight and 16 leaves were used for the quantification in *proATML1-nls-3xGFP* and the *atml1-3* mutant, respectively.

### GUS staining

Leaves were submerged in 90% acetone on ice for 30 min and washed twice with sodium phosphate buffer. The leaves were incubated in the GUS reaction buffer [30 mM $Na_2HPO_4$, 20 mM $NaH_2PO_4$, 1.5 mM $K_4Fe$, 1.5 mM $K_3Fe$, 500 mg/L X-Gluc, 0.1% Triton] at room temperature under vacuum for 1 h. The samples were further incubated at 37 °C until enough GUS signals were detected. After the incubation, the leaves were fixed in 4% paraformaldehyde in PBS at 4 °C overnight and washed twice with PBS. The samples were cleared with the ClearSee solution supplemented with 1 μl/ml SR2200.

### Reporting summary

Further information on research design is available in the Nature Portfolio Reporting Summary linked to this article.

## Data availability

All the data that support the findings of this study are included in this article and supplementary figures. Source data are provided with this paper.

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

## Acknowledgements

We are grateful to Hirofumi Ohmori (Osaka University, Japan) for DNA sequencing, Prof. Dolf Weijers (Wageningen University, The Netherlands) for *R2D2* seeds, the Arabidopsis Biological Research Center for *coi1* seeds and the Nottingham Arabidopsis Stock Centre for *atml1-3* and *35S-GFP-TUA6* seeds. We greatly thank Prof. Tatsuo Kakimoto (Osaka University, Japan) for supporting our project in his lab. We also thank the past and present members of the plant growth and development lab for their helpful comments and discussion. This work was supported by grants from the Japan Society for the Promotion of Science [16J00702 to H.I.; 20657012, 22687003, 23657036, 26440142, 18K06286 and 22K06280 to S.T.], EMBO Postdoctoral Fellowship [ALTF 128-2020 to H.I.] and the European Research Council [ERC-CoG CORKtheCAMBIA, agreement 819422 to A.P.M].

## Author contributions

H.I. and S.T. designed the experiments. H.I. performed the experiments and analysed the data. S.T. and G.J. provided transgenic plants. A.P.M provided the laboratory reagents and equipment. H.I., A.P.M., S.T., and G.J. wrote the manuscript. S.T. supervised the project. All authors discussed the results and commented on the manuscript.

## Competing interests

The authors declare no competing interests.
