## [Peer Review File · Nature Communications]

Epidermal injury-induced derepression of key regulator ATML1 in newly exposed cells elicits epidermis regenerationReviewer #1 (Remarks to the Author):

ATML1 is a well-established master regulator for epidermal identity. Its expression is largely restricted to the epidermal cells with a trace amount of transcription in the sub-epidermal mesophyll cells since embryogenesis. In this manuscript, the authors demonstrate that the mechanical property of the leaf epidermis determines the specificity of ATML1 expression. When leaf epidermal cells are surgically removed or damaged, the freshly exposed sub-epidermal mesophyll cells are released from mechanical constraint and start to express ATML1, which in turn leads to the conversion of sub-epidermal mesophyll cells to epidermal cells. By means of chemical inhibitors, the authors also show that cortical microtubule formation, MAPK signaling, and proteasome activity are required for the ectopic induction of ATML1. Together, this study provides an easily accessible model system for the dissection of position-dependent control of cell fate in plants, though the underlying mechanism remains unclear.

Questions/comments/suggestions:

1) "Many plant cells are totipotent"

Can the authors provide a list of plant cells that are totipotent?

2) "the molecular targets of positional information that are responsible for these cell fate changes have not been identified"

Is this a valid statement?

3) The study of 101x6-nls-3xGFP is not sufficient to reveal which regulatory sequences are sufficient for the ATML1 upregulation in mesophyll cells. Promoter deletion removing the 101-bp sequence is needed to verify its wound/mechanical injury inducibility.

4) Mesophyll is composed of two kinds of tissues: the upper palisade parenchyma and the lower spongy parenchyma. It is entirely unclear from the authors' study whether both of them are convertible and can be converted to upper adaxial and lower abaxial epidermal cells, respectively. The switch from abaxial to the adaxial epidermis in the last result section is somewhat abrupt. The authors are recommended to examine whether cell fate change (e.g. formation of trichomes) occurs in the upper palisade parenchyma below the damaged adaxial epidermis of 6-day-old seedlings. Additionally, previous studies have shown that ATML1 protein levels beyond the threshold, were necessary for the endo-reduplication and differentiation of Arabidopsis sepal cells into giant cells (Meyer et al., 2017). Since trichome cell differentiation necessitates endo-reduplication, it is also important to determine if ATML1 protein levels beyond the threshold, are necessary for the conversion of exposed palisade parenchyma cells into trichomes.

5) After damaging wider regions of the abaxial leaf epidermis, did the authors find the formation of only guard cells, or pavement cells as well? It is important to confirm this by analysis of markers, specific to guard cells and pavement cells.

6) The authors are recommended to test whether ATML1 and cell fate change can be induced in the other, undamaged mesophyll cell layer by damaging adaxial (or abaxial) epidermal cells + palisade (or spongy) parenchyma.

7) The functional significance of ATML1 in leaf epidermal regeneration was not studied. It is important to determine the efficiency of leaf epidermal regeneration in the atml1-3 null mutant. Moreover, it would also reveal if positive feedback regulation of ATML1 is necessary for leaf epidermal regeneration.

8) Fig. 1a: the expression in the sub-epidermal mesophyll cells is rather weak. An arrowhead could help readers easily locate the GFP positive cell.

9) Fig. 1g & i: Can the authors provide 3D images of a larger area to demonstrate the layer- and region-specific expression of ATML1 before and after wound/mechanical injury? Alternatively, sectioned image is needed to verify the specific expression of

ATML1 in the outermost position.

Reviewer #2 (Remarks to the Author):

The paper by Ida et al provides new insights into the mechanisms regulating cell fate specification, showing a direct connection between cell fate and cell position. The results are very interesting as expression of the epidermal cell identity regulator ATML1 appears to require physical cues or others which are directly integrated from the cell position, while previous cell fate cell positional mechanism were delivered by mobile endogenous cues such transcription factors or others. Some concerned should be clarified.

Major concerns

1) Can the authors explain how they know expression of ATML1 is not turned off/activated because the coverslip edge cut the leaf or killed cells (Figure 2. c, d, Suppl. Figure 4). Please include a picture showing leaf integrity in Suppl. Figure 4. In addition, it would highly recommendable to show if expression of GFP driven by other promoter such as 35S::GFP or other (101x6-nls-3xGFP?) is maintained in this procedure. The conditions used for the estradiol induction experiment (RPS5A>>GFP-ATML1) are not really the same (e.g. liquid medium would prevent damaged cells from drying or other), and therefore this control is not fully representative, although it strongly suggests that cells are most likely viable. Alternatively, the authors can repeat the induction experiment but using proATML1-nls-3xGFP in this occasion.

2) It is unclear, if epidermis can be regenerated on the abaxial side of leaf from a 9-day-old seedling or not. As expression of ATML1 is observed (Figure1), accumulation of ATML1 would be expected (is this the case?) thus leading to regeneration (please clarify). It is unclear why the regeneration ability of a leaf from a 9-day-old seedling is based on removal of a trichome (which should be on the adaxial side) when all previous experiments are apparently performed on the abaxial side. Furthermore, after concluding that the adaxial side is competent in 6-day-old seedlings based on ATML1 accumulation, regeneration is tested on the abaxial side. Please clarify if any type of regeneration is observed on the adaxial side. The observation of formation of a stomata in the inner tissues following a surgical injury is very interesting, please show in figure 5d the position of the adaxial/abaxial sides and indicate what the colored section represent, is it the hole left by the needle? This hole(?) cannot really be appreciated on the left panel of Figure 5d, please rotate it so this can be better appreciated. ATML1 expression or ATML1 accumulation needs to be shown in the inner cells in a similar experimental setting.

3) A proportion cannot be greater than 1. Please explain how this was calculated as the explanation in the methods section does not clarify how if "The proportion of GFP-positive cells was determined by calculating the number of GFP-positive mesophyll cells per total number of mesophyll cells in the observed field" the result can be greater than 1 in any case. Define more clearly what this parameter corresponds to in the panels of Figure 1m, n, o, Figure 2g, h, Suppl Figure 2), etc throughout the manuscript.

4) Do microtubule organization of mesophyll cells after removal of the epidermis eventually resembles that of the epidermis itself and what is the expectation? Can the authors add this control in Figure 3 by panels a-c? why do images b and j look different when they should be comparable based on procedure and quantification (panels c (3hr) and l (non-pressed))?

5) It looks like oryzalin inhibits ATML1 expression in the outer layer of a globular embryo(?) and PD98059 and MG132 also repress it in the outer layer (Suppl. Figure 5). Please provide a bright field or DIC image and indicate if these images correspond to the

dermatogen stage or which stage. Although ATML1 expression starts in the one-cell stage embryo, expression needs to be initiated in the daughter cells following each division (not necessarily be maintained as if these were the same cells undergoing maturation or differentiation), particularly in those divisions giving rise to the outer embryo tissues. Based on this idea and the expression patterns showed during embryogenesis, the statement "PD98059 and MG132 may be required for de-novo expression but not for the maintenance of ATML1 transcription" looks inaccurate, please rephrase/elaborate on this.

Other points

Figure 1. I) Please draw the contour rather than coloring cells.

"Many plant cells are totipotent..." They are not totipotent per se as they do not normally develop a whole organ. They can be reprogramed to acquire features related with toti- or pluripotency.

"which suggest...plant cell fate is mainly determined by the position of the cell rather than the lineage" The importance of the cell lineages as a requirement to integrate positional information has been shown in the Arabidopsis root. Please rephrase or remove.

"However, the molecular targets of positional information that are responsible for these cell fate changes have not been identified." The role of SHORT-ROOT, SCARECROW, PLETHORA3/5/7/1/2/4 as regulators of positional information in several regenerative process in the root has been shown. Please rephrase or remove.

"...suggesting that the epidermal cell death was sufficient to trigger ATML1 induction in the underlying mesophyll cells." Please use an alternative word to "death". If cell death is the cause, it is not sufficiently proven. Furthermore, some results in this paper appear to indicate that is not a signal coming from "cell death" that activates ATML1 expression.

"The high osmolality of the medium essentially mimicked the presence of overlying epidermal cells..." why? Please rephrase

Reviewer #3 (Remarks to the Author):

In this manuscript, Iida, Jürgens, and Takada propose that mechanical signals could play a role in restricting ATML1 expression to the epidermal layer. They show that upon epidermal removal ATML1 expression is activated in the newly exposed mesophyll, eliciting epidermal regeneration. De-repression of ATML1 in exposed mesophyll cells is suggested to require pressure release which indicates that mechanical pressure could be used to restrict ATML1 activity to the outermost cells to specify epidermal identity.

The manuscript is well written, and the results presented are generally of good quality.

The idea of mechanical feedback being involved in restricting the expression of genes involved in epidermis specifications is new and potentially very interesting. However, I have several major concerns regarding some experiments presented in this manuscript: 1. Pressure application experiments: Authors apply mechanical pressure on the leaf with removed abaxial epidermis and show that such treatment represses the ATML1 induction in the mesophyll cells. This experiment is performed by sandwiching the leaf between two coverslips and pressure them with the paperclip.

- How do authors control the pressure applied to the leaf? How the applied pressure compares to what could be generated by the epidermis in physiological conditions?
- The first true leaf of the 9 days-old plants is relatively big (nearly 1 cm in length – Fig 2B). Such a leaf has a well-specified midvein and does not have a homogeneous thickness (e.g. Mattsson et al., 2003, Fig 6D). The midvein (and likely secondary veins) region creates a bump on the surface, especially on the abaxial side, where authors

remove the epidermis. Additionally, it has fully developed trichomes on the adaxial surface. The leaf surface is therefore not flat. Pressing such a structure with two flat coverslips will likely lead to a strong mechanical compression at the midvein. How do authors know that the cells which respond are actually pressed? Unless this is shown (for example by confocal imaging – do the mesophyll cell still bulge or are they pressed by the coverslips?) one cannot know if the restriction of the ATML1 expression in a specific cell is caused by the mechanical pressure.

- To convincingly show the effect of the mechanical forces on the restriction of the ATML1 signal after the removal of the epidermis, a more controlled application of the pressure should be used. AFM or CFM should be used for such an experiment.

- What happens with the expression of the ATML1 in the epidermis if the intact leaf is pressed? If the ATML1 expression could be restricted by the mechanical pressure, one could expect that external application of the pressure on the intact leaf could reduce ATML1 expression in the epidermis.

2. Mannitol treatment: To test whether turgor pressure is involved in ATML1 derepression, authors grow plants in a liquid medium with low or high osmolality after removal of the epidermis and noticed that high osmolarity medium prevented the derepression of the ATML1 in the mesophyll. What is the control for this experiment? How can we be sure that this high concentration of the mannitol over 24 h is not simply perturbing the general functioning of the cells? The control experiment with another genetic marker should be included to exclude this possibility.

3. Wounding Response: Although authors investigate the potential wounding response after the removal of the epidermis by repeating their experiments in the background of coil1 mutant it cannot be excluded that JA independent wounding response still takes place.

4. Younger tissues: All experiments are done in relatively old leaves with differentiated spongy mesophyll. How does this compare with younger tissues (early leaf primordium, SAM)? From the mechanical perspectives, they may be very different (compact L2 layer in the SAM, air spaces in the mesophyll of the late leaf primordium). Is the ATML1 upregulated after ablation of the L1 cells at the SAM (e.g. Reinhard et al., 2003)?

5. Microtubules: What is the structure of the MT network in mesophyll cells before epidermis removal? How do the authors know if the structure of the MT network just after epidermis removal is normal and not perturbed by the recent epidermis peeling? Additionally, the MT network was observed 3 h after epidermis removal. In contrast, the ATML1 expression was observed one day after this operation. Why?

Dear Reviewers,

Thank you for reviewing our manuscript. We have revised the manuscript after carefully considering the reviewers' comments.

We appreciate the reviewers' comments and suggestions on our manuscript. They were very helpful to improve our manuscript. Above all, it is encouraging that the comments are essentially positive.

Detailed responses to comments from the individual reviewers are listed below.

Reviewer #1 (Remarks to the Author):

1) *"Many plant cells are totipotent"*

Can the authors provide a list of plant cells that are totipotent?

We are grateful to the Reviewer #1 for pointing out our incorrect statement. As suggested also by the Reviewer #2, many plant cells are not totipotent per se although they can acquire totipotency after reprogramming. We have changed the sentence as follows: "Many plant cells maintain their ability to be reprogrammed and develop into different cell types. Specifically, they can acquire totipotency after exogenous phytohormone- or injury-induced de-differentiation" (page 2 lines 28–30).

2) *"the molecular targets of positional information that are responsible for these cell fate changes have not been identified"*

Is this a valid statement?

We thank the Reviewer #1 to point this out. In this sentence, we meant that the molecular targets of positional information during mesophyll/epidermal cell fate changes have not been identified. In fact, several genes that possibly function downstream of positional signals during regeneration of root tissues have been already indicated (e.g., Marhava et al., 2019 Cell), as pointed out by the Reviewer #2. To avoid misleading the readers, we have removed this sentence, as the Reviewer #2 suggested.

3) *The study of 101x6-nls-3xGFP is not sufficient to reveal which regulatory sequences are sufficient for the ATML1 upregulation in mesophyll cells. Promoter deletion removing the 101-bp sequence is needed to verify its wound/mechanical injury inducibility.*

We agree with the Reviewer #1 that this experiment does not show which regulatory

sequences are sufficient for the *ATML1* upregulation in mesophyll cells. Our purpose of this experiment is to know whether the 101-bp region, which is sufficient for reporter gene expression in the epidermis of the embryos, is also sufficient for *ATML1* upregulation in mesophyll cells. Our results suggest that derepression of *ATML1* in the mesophyll requires a different or additional cis-regulatory sequence and that a positive feedback regulation via the ATML1-binding site (L1 box) in the 101-bp region plays little role during de-novo induction of *ATML1* after surgery. It would be an interesting future project to identify cis-regulatory sequences responsible for the derepression of *ATML1* in the mesophyll cells. We have changed the purpose sentence as follows: “Next, we examined whether known regulatory sequences are sufficient for the *ATML1* upregulation in mesophyll cells” (page 4 lines 94–95).

4) Mesophyll is composed of two kinds of tissues: the upper palisade parenchyma and the lower spongy parenchyma. It is entirely unclear from the authors' study whether both of them are convertible and can be converted to upper adaxial and lower abaxial epidermal cells, respectively. The switch from abaxial to the adaxial epidermis in the last result section is somewhat abrupt. The authors are recommended to examine whether cell fate change (e.g. formation of trichomes) occurs in the upper palisade parenchyma below the damaged adaxial epidermis of 6-day-old seedlings. Additionally, previous studies have shown that ATML1 protein levels beyond the threshold, were necessary for the endo-reduplication and differentiation of Arabidopsis sepal cells into giant cells (Meyer et al., 2017). Since trichome cell differentiation necessitates endo-reduplication, it is also important to determine if ATML1 protein levels beyond the threshold, are necessary for the conversion of exposed palisade parenchyma cells into trichomes.

We are grateful for the Reviewer #1 to raise this important point. As shown in the last result section, we injured adaxial trichome cells to easily trace the damaged epidermal region, and we detected ATML1 protein accumulation in the adaxial mesophyll cells underlying the damaged epidermis, implying a re-specification of adaxial mesophyll cell fate. To further trace the developmental fate of those *ATML1*-positive cells, we injured wider regions of the adaxial surface of the first and second leaves of 6-day-old seedlings and visualized trichome and guard cell identities. Our results revealed that *KATI*-positive guard cells and *GL2*-positive trichome-like cells were formed in the damaged region. These trichome-like cells were unusual in that they were less branched and were not surrounded by typical accessory cells, suggesting that they arise from an irregular developmental process. Furthermore, we also found that some *ATML1*-positive cells have plastid autofluorescence characteristic of the pavement cells. A few guard cells and

pavement cell-like cells in the damaged region were mislocated in the inner positions, possibly because of overlaying dysfunctional epidermal cells or disorganized proliferation of cells during the regeneration. In any case, these observations suggest that adaxial mesophyll cells have a potential to regenerate three epidermal cell types. These results were added in the revised manuscript (Supplementary Fig. 14).

As the Reviewer #1 pointed out, it has been reported that ATML1 protein accumulation above the threshold at G2/M phase leads to the endo-reduplication and generation of giant cells in sepals. However, it has not been shown whether it is also the case for leaf trichome development or not. Also, the purpose of the present study is to reveal molecular mechanisms that distinguish the outermost and inner cell fate, but not to reveal mechanisms that determine the developmental fate of different epidermal cell types such as pavement cells, guard cells, and trichomes. Therefore, the quantification of ATML1 protein levels during trichome regeneration is out of scope in the present study.

5) After damaging wider regions of the abaxial leaf epidermis, did the authors find the formation of only guard cells, or pavement cells as well? It is important to confirm this by analysis of markers, specific to guard cells and pavement cells.

We are grateful to the Reviewer #1 for raising this important point. To examine cell identities in more detail, we injured wider regions of the abaxial surface of the first and second leaves of 6-day-old seedlings. Epidermal cell death after the injury was confirmed by dead cell staining with propidium iodide and by disappearance of *proATML1-nls-3xGFP* signals. Five days after the injury, most of the leaf surface cells restored *proATML1-nls-3xGFP* expression, suggesting the regeneration of epidermal cells. To further assess cell identities acquired during the regeneration in the damaged region, we damaged wider regions of the adaxial leaf epidermis and performed histological analysis by using guard cell and trichome marker genes, as we mentioned above (comment 4). The pavement cell identity was assessed by *ATML1* expression and the presence of fewer and smaller plastids compared with those in the mesophyll cells. We found that guard cells, pavement cell-like cells and trichome-like cells were formed in the damaged region. These results are included in the revised manuscript (page 11 lines 273–290, pages 11–12 lines 294–304 and Supplementary Fig. 13, 14).

6) The authors are recommended to test whether ATML1 and cell fate change can be induced in the other, undamaged mesophyll cell layer by damaging adaxial (or abaxial) epidermal cells + palisade (or spongy) parenchyma.

We thank the Reviewer #1 for the suggestion. When we scratched the outer tissues of

young leaves with a needle, we most likely damaged the epidermis and mesophyll cells rather than damaging specifically the outermost epidermal layer because it is not feasible to damage wide regions of tissues precisely at one cell depth, with a needle manipulated by bare hands. Peeling off the outer tissues is a more specific way to remove mainly the epidermis although some mesophyll cells are also often removed. Thus, we are showing the *ATML1* induction and epidermis regeneration from mesophyll cells at the outermost position regardless of their origin (either L2 or L3).

7) The functional significance of ATML1 in leaf epidermal regeneration was not studied. It is important to determine the efficiency of leaf epidermal regeneration in the atml1-3 null mutant. Moreover, it would also reveal if positive feedback regulation of ATML1 is necessary for leaf epidermal regeneration.

We agree with the Reviewer #1 on this point. Five days after damaging the abaxial surface of the first or second leaf, we found *ATML1*-positive pavement cells in the damaged region of the wild-type leaf, suggesting that the epidermis was regenerated at the outer tissues. On the contrary, protruded cells that were distinct from the epidermis were found after the same treatment in *atml1-3* leaves. These protruded cells contained plastids that were larger than those of the intact pavement cells but smaller than those of mesophyll cells, which may imply that they have an intermediate cell identity between the epidermis and mesophyll cells. These results suggest that *ATML1* is required for the epidermis regeneration from the mesophyll cells after wounding. In addition to the surgery of the *atml1-3* mutant, we also examined the efficiency of regeneration in *RPS5A>>ATML1-SRDX* plants (Takada, 2013 PLOS ONE), in which expression of *ATML1* downstream genes and epidermal cell differentiation are suppressed after estradiol treatment. We found that induction of *ATML1-SRDX* inhibited the guard cell formation from the inner tissues of damaged leaves. These data are now included as Supplementary Fig. 13, 15.

8) Fig. 1a: the expression in the sub-epidermal mesophyll cells is rather weak. An arrowhead could help readers easily locate the GFP positive cell.

Thank you for the suggestion to improve Figure 1. We have added arrowheads in Fig. 1a and c as suggested.

9) Fig. 1g & i: Can the authors provide 3D images of a larger area to demonstrate the layer- and region-specific expression of ATML1 before and after wound/mechanical injury? Alternatively, sectioned image is needed to verify the specific expression of ATML1 in the outermost position.

Now we have added a sectioned image to clearly show that *ATML1* promoter activity was induced specifically in the outermost mesophyll cells (Supplementary Fig. 2).

Reviewer #2 (Remarks to the Author):

Major concerns

1) Can the authors explain how they know expression of ATML1 is not turned off/activated because the coverslip edge cut the leaf or killed cells (Figure 2. c, d, Suppl. Figure 4). Please include a picture showing leaf integrity in Suppl. Figure 4. In addition, it would highly recommendable to show if expression of GFP driven by other promoter such as 35S::GFP or other (101x6-nls-3xGFP?) is maintained in this procedure. The conditions used for the estradiol induction experiment (RPS5A>>GFP-ATML1) are not really the same (e.g. liquid medium would prevent damaged cells from drying or other), and therefore this control is not fully representative, although it strongly suggests that cells are most likely viable. Alternatively, the authors can repeat the induction experiment but using proATML1-nls-3xGFP in this occasion.

We appreciate this comment. As the Reviewer #2 suggested, we used another reporter line, in which a nuclear localized TdTomato was produced under the constitutive *RPS5A* promoter, for the press experiment. We did not find a significant change in TdTomato signals between pressed and non-pressed mesophyll cells, supporting the conclusion that cell viability and general transcription/translation machinery were not impaired by the pressure treatment.

We have added those data in Supplementary Fig. 8.

2) It is unclear, if epidermis can be regenerated on the abaxial side of leaf from a 9-day-old seedling or not. As expression of ATML1 is observed (Figure1), accumulation of ATML1 would be expected (is this the case?) thus leading to regeneration (please clarify). It is unclear why the regeneration ability of a leaf from a 9-day-old seedling is based on removal of a trichome (which should be on the adaxial side) when all previous experiments are apparently performed on the abaxial side. Furthermore, after concluding that the adaxial side is competent in 6-day-old seedlings based on ATML1 accumulation, regeneration is tested on the abaxial side. Please clarify if any type of regeneration is observed on the adaxial side. The observation of formation of a stomata in the inner tissues following a surgical injury is very interesting, please show in figure 5d the position of the adaxial/abaxial sides and indicate what the colored section represent, is it the hole left by the needle?

*This hole(?) cannot really be appreciated on the left panel of Figure 5d, please rotate it so this can be better appreciated. ATML1 expression or ATML1 accumulation **needs** to be shown in the inner cells in a similar experimental setting.*

We thank the Reviewer #2 for pointing out this issue. First, as we described in the original manuscript (lines 257–259), the epidermal cells were not regenerated from the mesophyll cells after removal of the abaxial leaf epidermis of 9-day-old seedlings, as judged by cell morphology. Although we did not include the data in the original manuscript, the accumulation of ATML1 protein was quite rarely detected in the exposed mesophyll cells of 10-day-old seedling one day after peeling of the abaxial leaf epidermis. This is consistent with the observation on the adaxial side: injury of adaxial trichome cells in 9-day-old seedlings did not cause ATML1 protein accumulation in the underlying mesophyll cells in one day. As we mentioned above (as a response to comment 4 from the Reviewer #1), we demonstrated the regeneration ability of mesophyll cells in the adaxial side of the leaves in the revised manuscript. After damaging wider regions of the adaxial surface of young leaves, we observed the formation of guard cells, ATML1-positive cells with small plastids and GL2-positive trichome-like cells in the damaged region. Some KATI-positive guard cells and ATML1-positive pavement cell-like cells were found in the inner positions, suggesting that they really derived from inner lineage cells. These observations suggest that at least three epidermal cell types were regenerated from the adaxial mesophyll cells. These results are included as Supplementary Fig. 14. We are grateful for the Reviewer#2's suggestions on Fig. 5. We have rotated the left model so that the colored region, which represents a region damaged by the needle, is clearly seen. Fig. 5c shows ATML1 expression at the cut surface of the 3D model in Fig. 5d. We described this information in the manuscript (page 11 lines 293–294).

3) A proportion cannot be greater than 1. Please explain how this was calculated as the explanation in the methods section does not clarify how if “The proportion of GFP-positive cells was determined by calculating the number of GFP-positive mesophyll cells per total number of mesophyll cells in the observed field” the result can be greater than 1 in any case. Define more clearly what this parameter corresponds to in the panels of Figure 1m, n, o, Figure 2g, h, Suppl Figure 2), etc throughout the manuscript.

Thank you for pointing out our misuse of the term “proportion”. The proportion was normalized in each data set: namely, the average proportion of GFP-positive cells in treated mesophyll cells was divided by the average proportion of those in control samples. Therefore, the calculated values can be greater than 1. As the Reviewer #2 suggested, we

should have described it as “relative” proportion of GFP-positive cells. We have changed the labels in the revised Figures.

4) Do microtubule organization of mesophyll cells after removal of the epidermis eventually resembles that of the epidermis itself and what is the expectation? Can the authors add this control in Figure 3 by panels a-c? why do images b and j look different when they should be comparable based on procedure and quantification (panels c (3har) and l (non-pressed))?

As suggested by the Reviewer #2, we have included data showing microtubule array patterns in the epidermis and mesophyll cells in the intact leaves as Supplementary Fig. 10a–c. Cortical microtubule arrays visualized by GFP-TUA6 were prominent in the anticlinal cell boundaries of the epidermis, whereas few cortical microtubules were observed in the anticlinal cell boundaries of inner mesophyll cells. Increase in number of cortical microtubule arrays in the exposed mesophyll cells after at 3 har led us to conclude that microtubule organization of exposed mesophyll cells resembled that of the epidermis rather than that of inner mesophyll cells. However, microtubule array patterns of exposed mesophyll cells were not completely same as those of the epidermis. This observation is consistent with the fact that mesophyll cells of 9-day-old seedlings that are suitable for the peeling experiments were not able to transdifferentiate into epidermal cells. We have added these data in Supplementary Fig. 10c,d. Please note that microtubule arrays were not stable after fixation and clearing, which made it difficult to observe further internal tissues and reconstruct 3D images. Therefore, we were not able to follow the changes in microtubule organization in the regenerating epidermal cells.

GFP signals in Fig. 3b appear weaker compared with those in Fig. 3j probably because CLSM laser power and fluorescence signal intensity change depending on the day; images 3b and 3j were obtained on different days and therefore cannot be compared. Quantification and comparison in each experiment have been done from the data taken on the same day, with the same setting.

5) It looks like oryzalin inhibits ATML1 expression in the outer layer of a globular embryo(?) and PD98059 and MG132 also repress it in the outer layer (Suppl. Figure 5). Please provide a bright field or DIC image and indicate if these images correspond to the dermatogen stage or which stage. Although ATML1 expression starts in the one-cell stage embryo, expression needs to be initiated in the daughter cells following each division (not necessarily be maintained as if these were the same cells undergoing maturation or differentiation), particularly in those divisions giving rise to the outer

embryo tissues. Based on this idea and the expression patterns showed during embryogenesis, the statement “PD98059 and MG132 may be required for de-novo expression but not for the maintenance of ATML1 transcription” looks inaccurate ,please rephrase/elaborate on this.

We are sorry for causing this confusion. Firstly, the oryzalin treated embryo in Supplementary Fig. 5 (Supplementary Fig. 11 in the revised manuscript) was at the eight-cell stage but not at the globular stage. The oryzalin treatment seems to inhibit cell divisions and arrest embryo development at early stages. Nonetheless, the *ATML1* expression was normal in this embryo; GFP signals were detected in all the cells. Secondly, although GFP signals in PD98059 and MG132-treated embryos appear weak in this figure, they are within the range of variation observed in the control embryos. As the reviewer suggested, we have added bright field images and other pictures showing variation of *ATML1* signals in the control and chemical-treated embryos. Last, we agree with the idea that developing embryos may require de-novo activation of *ATML1* after each cell division to maintain the outermost cell-specific expression of *ATML1*. If this is the case, it is difficult to distinguish between the ‘de-novo initiation’ and ‘maintenance’ of *ATML1* transcription. We assume that target components of oryzalin, PD98059 and MG132 may be required for the derepression of *ATML1* in the inner-lineage cells but not for the *ATML1* expression already established in the surface cells. Because *ATML1* is already expressed at the one-cell stage, treatment of oryzalin, PD98059 and MG132 at much earlier stages may be required for the inhibition of *ATML1* expression. We have rephrased the sentence as “target components of oryzalin, PD98059 and MG132 may be required for de-novo expression of *ATML1* during mesophyll cell re-specification but not for the already established expression of *ATML1* in the outermost cells”.

Other points

Figure 1. l) Please draw the contour rather than coloring cells.

We have drawn the contour as the Reviewer #2 suggested.

“Many plant cells are totipotent...” They are not totipotent per se as they do not normally develop a whole organ. They can be reprogramed to acquire features related with toti- or pluripotency.

We agree with the Reviewer #2 that this sentence is inaccurate. We have changed the sentence as follows: “Many plant cells maintain their ability to be reprogrammed and develop into different cell types. Specifically, they can acquire totipotency after exogenous phytohormone- or injury-induced de-differentiation.”

“which suggest...plant cell fate is mainly determined by the position of the cell rather than the lineage“ The importance of the cell lineages as a requirement to integrate positional information has been shown in the Arabidopsis root. Please rephrase or remove.

We agree with the Reviewer #2 that cell lineage-dependent cell fate decision is also known to play an important role in plant development. To avoid misleading the readers, we have changed the sentence as follows: “plant cell fate can be re-specified according to the position of the cell, regardless of its lineage, during regeneration”.

“However, the molecular targets of positional information that are responsible for these cell fate changes have not been identified.” The role of SHORT-ROOT, SCARECROW, PLETHORA3/5/7/1/2/4 as regulators of positional information in several regenerative process in the root has been shown. Please rephrase or remove.

We are sorry for causing the confusion. As we mentioned above (as a response to comment 2 from the Reviewer #1), we meant that the molecular targets of positional signals that direct the epidermal/mesophyll cell fate changes have not been identified yet. We have removed this sentence to avoid confusion.

“...suggesting that the epidermal cell death was sufficient to trigger ATML1 induction in the underlying mesophyll cells.” Please use an alternative word to "death". If cell death is the cause, it is not sufficiently proven. Furthermore, some results in this paper appear to indicate that is not a signal coming from “cell death” that activates ATML1 expression. Thank you for pointing out the inaccurate use of “death”. We have changed it to “cell injury” in the revised manuscript.

“The high osmolality of the medium essentially mimicked the presence of overlying epidermal cells...” why? Please rephrase

High osmolarity of the medium absorbs water from the cells, which inhibits cell expansion. The high osmolality of the medium mimicked the presence of overlying epidermal cells probably because inhibition of mesophyll cell expansion is involved in *ATML1* repression in the mesophyll cells. Consistent with this idea, mechanical pressure from the epidermis or coverslips inhibit cell expansion as well as *ATML1* expression in the mesophyll cells. To clarify the statement, we have added sentences as follows: “To test whether mesophyll cell expansion or high turgor pressure is required for *ATML1* derepression” and “The high osmolality of the medium, which absorbs water from the

cells and inhibits cell expansion, essentially mimicked the presence of overlying epidermal cells. Considering that the pressure treatment significantly decreased cell expansion as well as *ATML1* expression of exposed mesophyll cells, inhibition of cell expansion or plasma membrane stretching by the mechanical pressure from the epidermis may mediate *ATML1* repression in the inner mesophyll cells.”

Reviewer #3 (Remarks to the Author):

1. Pressure application experiments: Authors apply mechanical pressure on the leaf with removed abaxial epidermis and show that such treatment represses the ATML1 induction in the mesophyll cells. This experiment is performed by sandwiching the leaf between two coverslips and pressure them with the paperclip.

- *How do authors control the pressure applied to the leaf? How the applied pressure compares to what could be generated by the epidermis in physiological conditions?*

We thank the Reviewer #3 to point out these important issues. We controlled the pressure by adjusting the paperclip; we adjusted the pressure so that some water came out from the mesophyll cells when we slightly touched the coverslips. It would be quite challenging to predict and measure the mechanical environment of mesophyll cells because their shapes are not uniform. However, judging by the length of the pressed mesophyll cells, which was comparable to that of intact mesophyll cells overlaid by the epidermis, we concluded that the coverslip method successfully applied appropriate pressure to the exposed mesophyll cells at similar levels as imposed by the epidermis of the intact leaves (Supplementary Fig. 6).

- *The first true leaf of the 9 days-old plants is relatively big (nearly 1 cm in length – Fig 2B). Such a leaf has a well-specified midvein and does not have a homogeneous thickness (e.g. Mattsson et al., 2003, Fig 6D). The midvein (and likely secondary veins) region creates a bump on the surface, especially on the abaxial side, where authors remove the epidermis. Additionally, it has fully developed trichomes on the adaxial surface. The leaf surface is therefore not flat. Pressing such a structure with two flat coverslips will likely lead to a strong mechanical compression at the midvein. How do authors know that the cells which respond are actually pressed? Unless this is shown (for example by confocal imaging – do the mesophyll cell still bulge or are they pressed by the coverslips?) one cannot know if the restriction of the ATML1 expression in a specific cell is caused by the mechanical pressure.*

We thank the Reviewer #3 for raising these valid concerns. First of all, we always pressed the distal region of the leaves where the mid vein developed at a lesser extent compared

with that in the proximal region. Therefore, the mid vein was not likely to affect the experiments. Nonetheless, we agree with the Reviewer #3 that we should demonstrate that exposed mesophyll cells were really pressed by the coverslips. To this end, we carefully observed histological sections of mesophyll cells with the pressure treatment and measured the cell length of mesophyll cells along the dorsoventral axis. We found that the cell length of outer mesophyll cells in the pressed region was not significantly different from that of mesophyll cells overlaid with the epidermis in non-pressed region. This result suggests that the coverslip/paperclip conferred appropriate pressure to the mesophyll cells. We have included these data as Supplementary Fig. 6.

- *To convincingly show the effect of the mechanical forces on the restriction of the ATML1 signal after the removal of the epidermis, a more controlled application of the pressure should be used. AFM or CFM should be used for such an experiment.*

We agree with the Reviewer #3 that the use of AFM or CFM would enable more refined tests of the effect of the mechanical force on *ATML1* expression, if applicable. Unfortunately, it would not be technically feasible to continuously apply mechanical force for 24 hours on peeled leaves by using AFM/CFM; the leaves will dry out. As we mentioned above, the paper clip provided enough pressure to reduce the expansion of the mesophyll cells and mimicked the mesophyll cell length of the intact leaves. As a pioneer study, we have provided the first evidence for the connection between epidermal/mesophyll cell fate decision and mechanical forces. Further studies should include more refined analysis and measurement, which are beyond the scope of the present study. Also, we believe that our paper clip/coverslip method will provide a novel opportunity to study the effect of mechanical forces on gene expression even if expensive sets of equipment are not available, which will contribute to the further development of this research area.

- *What happens with the expression of the ATML1 in the epidermis if the intact leaf is pressed? If the ATML1 expression could be restricted by the mechanical pressure, one could expect that external application of the pressure on the intact leaf could reduce ATML1 expression in the epidermis.*

We thank the Reviewer #3 for the interest and comment on the possible effect of the mechanical pressure on differentiated epidermal cells. When we pressed the intact leaves, the leaf epidermis still showed the same level of *ATML1* expression as seen in the non-pressed region. We assume that the mechanical pressure can efficiently repress only the de-novo expression of *ATML1* during epidermal regeneration; once *ATML1* expression is

established above a certain threshold, positive feedback regulation through the L1 box may maintain *ATML1* expression even in the presence of the inhibitory mechanical pressure. These data are added as Supplementary Fig. 9.

2. Mannitol treatment: To test whether turgor pressure is involved in ATML1 derepression, authors grow plants in a liquid medium with low or high osmolality after removal of the epidermis and noticed that high osmolarity medium prevented the derepression of the ATML1 in the mesophyll. What is the control for this experiment? How can we be sure that this high concentration of the mannitol over 24 h is not simply perturbing the general functioning of the cells? The control experiment with another genetic marker should be included to exclude this possibility.

We acknowledge the Reviewer#3's concern on the integrity of leaf cells after the mannitol treatment. To test whether or not the general function of the cells (i.e., transcriptional and translational machineries) was perturbed by the high concentration of mannitol, we did the same mannitol experiment using *GFP-ATML1* overexpressing plants (*RPS5A>>GFP-ATML1*). We confirmed that *GFP-ATML1* signals were induced normally in the exposed mesophyll cells and were not affected by the mannitol treatment. We have added these data as Supplementary Fig. 7c,d.

*3. Wounding Response: Although authors investigate the potential wounding response after the removal of the epidermis by repeating their experiments in the background of *coi1* mutant it cannot be excluded that JA independent wounding response still takes place.*

We agree with the Reviewer #3 that JA-independent wounding response may be involved in *ATML1* derepression. To clarify this point, we have added the following sentence in the revised manuscript: "although we cannot exclude the possibility that JA-independent wounding response might be involved in the *ATML1* induction" (page 6 lines 140–142).

4. Younger tissues: All experiments are done in relatively old leaves with differentiated spongy mesophyll. How does this compare with younger tissues (early leaf primordium, SAM)? From the mechanical perspectives, they may be very different (compact L2 layer in the SAM, air spaces in the mesophyll of the late leaf primordium). Is the ATML1 upregulated after ablation of the L1 cells at the SAM (e.g. Reinhard et al., 2003)?

We thank the Reviewer #3 for raising the possibility that age-dependent tissue organization changes may alter the mechanical environment and hence *ATML1* derepression efficiency in the mesophyll cells. To test this possibility, we scratched the

shoot apical region of 4-day-old seedlings by using a needle and observed *ATML1* expression after one day. We found an upregulation of *ATML1* expression in the L2 layer of leaf primordia cells underneath the damaged epidermis, suggesting that the compact L2 layer cells of younger tissues can still derepress *ATML1* expression in response to the injury of L1 layer cells. We have added these data as Supplementary Fig. 3.

5. Microtubules: What is the structure of the MT network in mesophyll cells before epidermis removal? How do the authors know if the structure of the MT network just after epidermis removal is normal and not perturbed by the recent epidermis peeling? Additionally, the MT network was observed 3 h after epidermis removal. In contrast, the ATML1 expression was observed one day after this operation. Why?

We are grateful to the Reviewer #3 for the useful comments that the microtubule network in the exposed mesophyll cells observed right after removal of the epidermis may be different from that in the intact mesophyll cells.

GFP-TUA6 signals were unstable after fixation and clearing, which made it difficult to observe internal tissues of mature leaves without peeling off the epidermis. Therefore, we utilized younger leaves to observe microtubules in the intact mesophyll cells without removal of the epidermis or clearing treatment. We found that cortical microtubule arrays were prominent in the anticlinal cell boundaries of the epidermis, whereas few microtubules were observed in the anticlinal cell boundaries of inner mesophyll cells, suggesting that the microtubule network in the intact mesophyll cells is similar to that observed in the exposed mesophyll cells right after removal of the epidermis. We have added these data as Supplementary Fig. 10 in the revised manuscript.

We assume that there are two possible reasons that can explain the time delay between the microtubule network formation and the *ATML1* expression. 1) GFP-TUA6 protein should be present in the cytosol of mesophyll cells even before removal of the epidermis and is therefore ready to be assembled upon induction, whereas the nls-3xGFP reporter should be transcribed and translated in the mesophyll cells only after the *ATML1* promoter is activated, which may cause a delay in the detection of the response. 2) Although cortical microtubule formation at the plasma membrane was required for *ATML1* derepression in the exposed mesophyll cells, the cortical microtubule itself cannot directly activate gene expression in the nuclei. The time delay may suggest the existence of signal transduction pathways to convey the signal from the plasma membrane to the nucleus.

Reviewer #1 (Remarks to the Author):

The authors have satisfactorily responded to all my questions and made the necessary changes to the manuscript. I have no additional comments and recommend accepting the paper.

Reviewer #2 (Remarks to the Author):

The authors have addressed all the concerns raised by this reviewer. As detailed below a couple of recommendations are made.

Concern 1: The authors demonstrate that the observed repression of ATML1 is not caused by the manipulation procedure. It would be recommendable to include images in Suppl. Figure 8 in addition to the quantifications.

Concern 2: The authors demonstrate regeneration on the leaf adaxial side as shown in Suppl. Figure 14, particularly in panels a) and b). It is unclear if the cell pointed by the arrow in panel c) is indeed a mesophyll cell as it appears to be on the edge. Nonetheless, it is up to the authors to leave or remove this experiment in c) as regeneration is demonstrated by previous panels a) and b) already. ATML expression in inner cells has been clarified to be already shown in Figure 5c. The 3D models are much better arranged now and fulfill their purpose.

Concern 3: The authors have clarified the use of this parameter in the manuscript.

Concern 4: Although it is unclear why there is not a maximum projection image for the subepidermal mesophyll cells in the case in which the epidermis was removed (Suppl. Figure 10b) so it can be compared with the case in which the epidermis was removed (Suppl. Figure 10d), these data support the expectation raised by the authors. Nonetheless, I strongly recommend adding the mentioned maximum projection image.

Concern 5: This reviewer agrees with the current interpretation given by the authors based on the experiments shown.

Other points:

"Figure 1.1": It looks much clearer now

"Many plant cells are totipotent...": The new sentences are much more accurate and read well, although one may argue about the existence of endogenous reprogramming signals.

"which suggest...plant cell fate is mainly determined by the position of the cell rather than the lineage": the new sentence is fair.

"However, the molecular targets of positional information that are... Please rephrase or remove": The sentence has been removed.

"...suggesting that the epidermal cell death was sufficient to trigger ATML1 induction in the underlying mesophyll cells... Please use an alternative word to `death`": The word `death` has been changed to injury which is fair.

"The high osmolality....why? Please rephrase": This idea has been clarified and new explanatory sentences added.

Reviewer #3 (Remarks to the Author):

"We controlled the pressure by adjusting the paperclip; we adjusted the pressure so that some water came out from the mesophyll cells when we slightly touched the coverslips. It would be

quite challenging to predict and measure the mechanical environment of mesophyll cells because their shapes are not uniform. However, judging by the length of the pressed mesophyll cells, which was comparable to that of intact mesophyll cells overlaid by the epidermis, we concluded that the coverslip method successfully applied appropriate pressure to the exposed mesophyll cells at similar levels as imposed by the epidermis of the intact leaves (Supplementary Fig. 6)."

The paper clip cannot be properly controlled to apply comparable pressure. What does it mean: "... so that some water came out from the mesophyll cells when we slightly touched the coverslips."? How this can be reproduced by other scientists? Each clip will differ from each other, each leaf will be different... The authors state that it is challenging to predict and measure pressure in this system as the mesophyll cell shapes are non-uniform. Not only cell shapes are non-uniform, but the leaf surface is also non-uniform, and there are trichomes that stick out from the surface. Even if they were able to apply the pressure of a given force, they would not know how this pressure is distributed over the pressed surface. Considering all these variables how do authors select the cells that are really pressed? How do they know that this pressure is comparable to that exerted by the epidermis? In my opinion, this critical experiment in its current setup doesn't convincingly show that ML1 in the subepidermal layers is repressed by mechanical forces and that the epidermis is exerting a mechanical pressure on the mesophyll to prevent the ML1 expression in the mesophyll.

"We always pressed the distal region of the leaves where the midvein developed at a lesser extent compared with that in the proximal region. Therefore, the midvein was not likely to affect the experiments."

I somehow disagree. Even if the leaf surface geometry would be perfectly flat, the density of the internal tissue would differ between the regions located at the veins and between them (absence and presence of air spaces). So even if the pressure would be homogenous and the surface would be perfectly flat the pressure would likely vary. Trichomes, which are very big cells located at the surface of the leaf, and whose base is known to protrude from the leaf surface, are expected to further increase this heterogeneity.

"... we scratched the shoot apical region of 4-day-old seedlings by using a needle and observed ATML1 expression after one day. We found an upregulation of ATML1 expression in the L2 layer of leaf primordia cells underneath the damaged epidermis, suggesting that the compact L2 layer cells of younger tissues can still derepress ATML1 expression in response to the injury of L1 layer cells. We have added these data as Supplementary Fig. 3."

Indeed, one can observe some mild expression of GFP in the inner layers of the young leaf after ablation of the epidermis (Fig. S3C). However, the GFP signal can also be observed in the inner tissues at not ablated locations (right side of the leaf, under the leaf margin in Fig. S3C). How this can be explained? There is no ablation at this location, but the inner tissue GFP signal seems to be even stronger compared to the ablated areas? Did the authors consider the possibility of pML1 not always being specific to the L1 layer (Fig. S1C in <https://doi.org/10.1016/j.devcel.2013.08.017>)? Also, why entire ablated epidermis isn't shown in S3C?

"... we utilized younger leaves to observe microtubules in the intact mesophyll cells without the removal of the epidermis or clearing treatment. We found that cortical microtubule arrays were prominent in the anticlinal cell boundaries of the epidermis, whereas few microtubules were observed in the anticlinal cell boundaries of inner mesophyll cells, suggesting that the microtubule network in the intact mesophyll cells is similar to that observed in the exposed mesophyll cells right after removal of the epidermis."

As the authors do not compare the intact mesophyll cells and exposed mesophyll cells right after removal of the epidermis at the same developmental stages of the leaf they do not demonstrate that the CMT network is not affected directly after epidermal removal. The same leaf developmental stages would need to be compared.

Dear Reviewers,

Thank you for reviewing our manuscript. We are happy to learn that Reviewer 1 is now satisfied with our manuscript and that Reviewer 2 has few concerns except for presentation of two figures. We have revised the manuscript after carefully considering the reviewers' comments. Detailed responses to comments from the individual reviewers are listed below.

Reviewer #1 (Remarks to the Author):

The authors have satisfactorily responded to all my questions and made the necessary changes to the manuscript. I have no additional comments and recommend accepting the paper.

We thank Reviewer 1 for recommending the acceptance of our manuscript.

Reviewer #2 (Remarks to the Author):

Concern 1: The authors demonstrate that the observed repression of ATML1 is not caused by the manipulation procedure. It would be recommendable to include images in Suppl. Figure 8 in addition to the quantifications.

We are grateful for the suggestion. The images have been added to Suppl. Figure 8, as suggested.

Concern 2: The authors demonstrate regeneration on the leaf adaxial side as shown in Suppl. Figure 14, particularly in panels a) and b). It is unclear if the cell pointed by the arrow in panel c) is indeed a mesophyll cell as it appears to be on the edge. Nonetheless, it is up to the authors to leave or remove this experiment in c) as regeneration is demonstrated by previous panels a) and b) already. ATML expression in inner cells has been clarified to be already shown in Figure 5c. The 3D models are much better arranged now and fulfill their purpose.

We thank Reviewer 2 for the concern. Although this arrowed cell appears to be on the edge, the cell was indeed located in the inner position (confirmed by z-stack images). We decided to retain this image to show that not only guard cells and trichomes but also pavement cells were regenerated after the injury.

Concern 4: Although it is unclear why there is not a maximum projection image for the subepidermal mesophyll cells in the case in which the epidermis was removed (Suppl. Figure 10b) so it can be compared with the case in which the epidermis was removed (Suppl. Figure 10d), these data support the expectation raised by the authors. Nonetheless, I strongly recommend adding the mentioned maximum projection image.

We thank Reviewer 2 for the recommendation. Suppl. Figure 10b is not shown as a maximum projection image because the purpose of Suppl. Figure 10a and 10b is to compare cortical microtubule arrays between epidermal and mesophyll tissues in an intact leaf. Therefore, Suppl. Figure 10b is supposed to be compared with Suppl. Figure 10a, which is not a maximum projection image.

Please note that mesophyll cells in Suppl. Figure 10b are overlaid with the epidermis. Therefore, cortical microtubules in the epidermal layer were also unavoidably projected in the maximum projection image. Because such an image would be misleading, we did not add the maximum projection image.

Reviewer #3 (Remarks to the Author):

1) *“We controlled the pressure by adjusting the paperclip; we adjusted the pressure so that some water came out from the mesophyll cells when we slightly touched the coverslips. It would be quite challenging to predict and measure the mechanical environment of mesophyll cells because their shapes are not uniform. However, judging by the length of the pressed mesophyll cells, which was comparable to that of intact mesophyll cells overlaid by the epidermis, we concluded that the coverslip method successfully applied appropriate pressure to the exposed mesophyll cells at similar levels as imposed by the epidermis of the intact leaves (Supplementary Fig. 6).”*

The paper clip cannot be properly controlled to apply comparable pressure. What does it mean: “... so that some water came out from the mesophyll cells when we slightly touched the coverslips.”? How this can be reproduced by other scientists? Each clip will differ from each other, each leaf will be different...

We are grateful for the comments. We did not include the sentence ‘we adjusted the pressure so that some water came out from the mesophyll cells when we slightly touched the coverslips’ in the Method section because this information is somewhat subjective, as Reviewer 3 suggested. This paper clip experiment can be reproduced by using the mesophyll cell length as a measure of the pressure on the mesophyll cells. Namely, one can apply a pressure so that mesophyll cell length becomes comparable to that of intact mesophyll cells overlaid by the epidermis. Please note that even apparently unstable pressures exerted by paper clips consistently repressed cell expansion and *ATML1* induction, suggesting that mesophyll cells response to mechanical pressure is robust (please see discussion below). This fact suggests that practically it was not necessary to exactly control the pressure level to examine the effect on *ATML1* expression. Also, in any cases, it is not technically possible to exactly predict and apply the same level of pressure as exerted by the epidermis, as Reviewer 3 pointed out below. Therefore, the paper clip method is good enough for our purpose.

2) *The authors state that it is challenging to predict and measure pressure in this system as the mesophyll cell shapes are non-uniform. Not only cell shapes are non-uniform, but the leaf surface is also non-uniform, and there are trichomes that stick out from the surface. Even if they were able to apply the pressure of a given force, they would not know how this pressure is distributed over the pressed surface. Considering all these variables how do authors select the cells that are really pressed? How do they know that this pressure is comparable to that exerted by the epidermis?*

We thank Reviewer 3 for the concerns. As we mentioned in the previous version of the manuscript, we used mesophyll cell length to know that the pressure was really applied or released in these cells. Cell length measurement (Suppl. Figures 5 and 6) has shown that cell length is increased in exposed mesophyll cells, suggesting a release from the pressure exerted by the epidermis. By contrast, when the exposed mesophyll cells were pressed with a paper clip, the length of those mesophyll cells became comparable to that of the intact subepidermal mesophyll cells. Therefore, from cell length measurement, we can know that the paper clips provided enough pressure to inhibit cell expansion of exposed mesophyll cells to a similar extent as that of intact mesophyll cells. The consistent and overall decrease in mesophyll cell length by pressure treatment (Suppl. Figure 6) was well correlated with the almost complete absence of *ATML1* induction (Figure 2g). Moreover, in the revised manuscript, we have shown a positive correlation between pressure level (predicted by cell expansion) and *ATML1* repression by distinguishing “pressed” and “non-pressed” cells (please see below).

3) *In my opinion, this critical experiment in its current setup doesn't convincingly show*

that ML1 in the subepidermal layers is repressed by mechanical forces and that the epidermis is exerting a mechanical pressure on the mesophyll to prevent the ML1 expression in the mesophyll.

We have already shown two lines of evidence to support this conclusion in the previous version of the manuscript. (i) Exposed mesophyll cells expanded outward when the overlying epidermis was removed, suggesting that the epidermis exerts a mechanical pressure on subepidermal mesophyll cells. Please note that cell shape changes have been used previously to predict the direction of forces within complex plant tissues (Vermeer et al 2014 Science, Hoermayer et al 2020 PNAS). (ii) When the cell expansion of exposed mesophyll cells was prevented by the mannitol treatment or pressure treatment with a paper clip, *ATML1* upregulation was repressed in those mesophyll cells. This result suggests that mesophyll cell expansion, triggered by the removal of the epidermis, is required for de-novo *ATML1* expression.

We believe these results show that mesophyll cells are repressed by the pressure exerted by the epidermis and that this pressure inhibits mesophyll cell expansion as well as de-novo *ATML1* expression. Nonetheless, to further prove the validity of our conclusion, we have included additional experimental evidence of a positive correlation between mechanical de-repression and *ATML1* induction in the current version of the manuscript. We focused on “boundary mesophyll cells”, which are located near the boundary between peeled/unpeeled regions and are still overlaid by the epidermis. Because these mesophyll cells are expected to be under different levels of pressures from partially damaged epidermal layer, we used these cells to examine the correlation between cell expansion (namely pressure level) and *ATML1* induction. Indeed, some boundary mesophyll cells overlaid with the epidermis showed invasion toward the epidermal layer, suggesting a partial release from the pressure. We found that most of the invaded mesophyll cells showed *ATML1* induction, whereas few uninvaded cells did. We have included these data as Suppl. Figure 7 and combined original Suppl. Figure 7, 8, and two additional images into one figure (Suppl. Figure 8).

Altogether, these results are consistent with the conclusion that mechanical de-repression is responsible for *ATML1* induction during epidermis regeneration. Nevertheless, our data do not exclude the possibility that other stimuli caused by the wounding may also contribute to the *ATML1* de-repression. Therefore, to avoid the impression that *ATML1* repression solely depends on the mechanical pressure, we have changed the title, rephrased misleading sentences, and mentioned other possible scenarios. These changes are highlighted in the text.

The observed robustness of mesophyll cell response to mechanical pressure is consistent with the expectation that the epidermis should exert variable pressure on mesophyll cells: not only cell shape and leaf structure but also the environment could affect the pressure from the epidermis (e.g., wind, rain, drought, and insect visit). Therefore, it is reasonable to assume that mesophyll cells are programmed to respond to a wide range of pressures from the epidermis to achieve a stable output (i.e. inhibition of epidermal cell differentiation in the inner cells).

4) “*We always pressed the distal region of the leaves where the midvein developed at a lesser extent compared with that in the proximal region. Therefore, the midvein was not likely to affect the experiments.*”

I somehow disagree. Even if the leaf surface geometry would be perfectly flat, the density of the internal tissue would differ between the regions located at the veins and between them (absence and presence of air spaces). So even if the pressure would be homogenous and the surface would be perfectly flat the pressure would likely vary. Trichomes, which are very big cells located at the surface of the leaf, and whose base is known to protrude from the leaf surface, are expected to further increase this heterogeneity.

We are grateful for the comment. We agree with Reviewer 3 that because of the heterogeneity of leaf tissues, it is not technically possible to apply the same level of pressure as exerted by the epidermis on mesophyll cells. However, even in such heterogeneous tissues, we obtained consistent

results with our experimental setup, in terms of cell expansion and *ATML1* expression. This fact again indicates that mesophyll cells show a robust response to mechanical pressure: they can respond to a wide range of pressures to inhibit cell expansion and *ATML1* induction. In other words, the performed paper clip method was able to apply a sufficient range of pressure on mesophyll cells to inhibit *ATML1* de-repression. Therefore, our paper clip method is valid and sufficient for our purpose.

5) “... we scratched the shoot apical region of 4-day-old seedlings by using a needle and observed *ATML1* expression after one day. We found an upregulation of *ATML1* expression in the L2 layer of leaf primordia cells underneath the damaged epidermis, suggesting that the compact L2 layer cells of younger tissues can still derepress *ATML1* expression in response to the injury of L1 layer cells. We have added these data as Supplementary Fig. 3.”

Indeed, one can observe some mild expression of GFP in the inner layers of the young leaf after ablation of the epidermis (Fig. S3C). However, the GFP signal can also be observed in the inner tissues at not ablated locations (right side of the leaf, under the leaf margin in Fig. S3C). How this can be explained? There is no ablation at this location, but the inner tissue GFP signal seems to be even stronger compared to the ablated areas? Did the authors consider the possibility of pM1 not always being specific to the L1 layer (Fig. S1C in <https://doi.org/10.1016/j.devcel.2013.08.017>)? Also, why entire ablated epidermis isn't shown in S3C?

We thank Reviewer 3 for the concerns. The GFP-positive mesophyll cell was located at the surface of the primordium as confirmed by a different optical section (please see the image below). Therefore, this cell was displaced to the surface after the injury of overlying epidermal layer (white arrowhead), and this GFP signal is still consistent with the idea that epidermal cell injury induces *ATML1* reporter expression in subepidermal cells of young leaf primordia. Also, we have already shown that weak *ATML1* expression is present in L2 layer of leaves (Figure 1a,c). Therefore, after the injury, we evaluated the “increase” of *ATML1* expression in comparison with the control leaves, as described in the manuscript.

Suppl. Figure 3c is an optical cross section reconstructed from a z-stack image series, starting from the plane where the injured epidermis was clearly seen. This is the reason why the entire ablated epidermis (including surface cell walls) is not shown in Suppl. Figure 3c although all the injured cells are visible, at least in part, and indicated with the red arrowhead. Please see Suppl. Figure 3a,b to confirm the entire region of the ablated epidermis.

6) “... we utilized younger leaves to observe microtubules in the intact mesophyll cells without the removal of the epidermis or clearing treatment. We found that cortical microtubule arrays were prominent in the anticlinal cell boundaries of the epidermis, whereas few microtubules were observed in the anticlinal cell boundaries of inner mesophyll cells, suggesting that the microtubule network in the intact mesophyll cells is similar to that observed in the exposed mesophyll cells right after removal of the epidermis.”

As the authors do not compare the intact mesophyll cells and exposed mesophyll cells right after removal of the epidermis at the same developmental stages of the leaf they do not demonstrate that the CMT network is not affected directly after epidermal removal. The same leaf developmental stages would need to be compared.

We are grateful for the concern. This experiment has been done to address the concern raised by Reviewer 3 *‘What is the structure of the MT network in mesophyll cells before epidermis removal? How do the authors know if the structure of the MT network just after epidermis removal is normal and not perturbed by the recent epidermis peeling?’*.

By utilizing young leaves of 6-day-old plants, we compared cortical microtubule (CMT) formation between epidermal and mesophyll cells in intact leaves without epidermis removal. We confirmed that mesophyll cells of the control leaves rarely formed CMTs, suggesting that the removal of the epidermis was not the cause of the CMT reduction in mesophyll cells at 0 dar. We believe that, as Reviewer 2 mentioned, this result supports our hypothesis that exposed mesophyll cells are more similar to the epidermis rather than intact mesophyll cells, in terms of the CMT formation.

As suggested by Reviewer 3, we cannot exclude the possibility that CMTs are formed in mesophyll cells of 9-day-old seedlings but not in 6-day-old seedlings and CMTs are decreased upon the removal of the epidermis in leaves of 9-day-old seedlings. As we explained in the previous response letter, GFP-TUA6 signals were unstable after fixation and clearing, which made it difficult to observe internal mesophyll tissues of thick leaves in 9-day-old seedlings without peeling off the epidermis. This is the reason why we utilized younger thin leaves of 6-day-old seedlings to compare CMT formation between epidermal and mesophyll cells “before epidermis removal”. To avoid the misunderstanding, we rephrased the relevant sentence to “in regard to the cortical microtubule density, the exposed mesophyll cells resembled the intact epidermal cells more than inner mesophyll cells of untreated younger leaves”

Please note that our main claim regarding CMTs is that CMT formation in exposed mesophyll cells is required for de-repression of *ATML1* during regeneration (Figure 3d–i). This paper describes the mechanisms underlying de-novo epidermal cell specification during regeneration: epidermal cell specification during regeneration does not necessarily follow the same mechanism as that used during normal developmental processes. Therefore, it is not relevant to our conclusions to see whether CMT arrays in exposed mesophyll cells resembled those of intact epidermal cells or CMTs of mesophyll cells were disrupted after removal of the epidermis. In fact, CMT arrays in exposed mesophyll cells at 1 dar were not completely the same as those in intact epidermal cells (please see Figure 3 and Suppl. Figure 10).